# Hypercone Assisted Contour Generation for Out-Of-Distribution Detection

## Abstract

Recent advances in the field of out-of-distribution (OOD) detection have placed great emphasis on learning better representations suited to this task. While there have been distance-based approaches, distributional awareness has seldom been exploited for better performance. We present $HAC_k$-OOD, a novel OOD detection method that makes no distributional assumption about the data, but automatically adapts to its distribution. Specifically, $HAC_k$-OOD constructs a set of hypercones by maximizing the angular distance to neighbors in a given data-point's vicinity to approximate the contour within which in-distribution (ID) data-points lie. Experimental results show state-of-the-art FPR@95 and AUROC performance on *Near-OOD detection* and on *Far-OOD detection* on the challenging CIFAR-100 benchmark without explicitly training for OOD performance.

## 1 Introduction

Machine learning models are trained on a particular set of data, called the in-distribution (ID) set. During inference, it is possible that the trained model will receive samples drawn from a different distribution to the one it has been trained on. These observations are said to be out-of-distribution (OOD). It is important to be able to distinguish between ID and OOD instances for safe model deployment.

Existing OOD detection methods can be broadly categorized into training-based methods and post-processing distance-based methods Yang et al. (2024). Training-based methods aim to incorporate OOD detection capabilities directly into the model via train-time regularization Lu et al. (2024); Ming et al. (2022a). These methods typically modify the objective function or architecture to enhance sensitivity to OOD inputs, e.g., using auxiliary classifiers or network branches targeted at OOD detection Papadopoulos et al. (2021); Mohseni et al. (2020), adversarial training Yi et al. (2021); Chen et al. (2021), or self-supervised learning objectives Sehwag et al. (2021); Mohseni et al. (2020); Hendrycks et al. (2019a); Sun et al. (2022a). They may also take a two-step modeling approach Liang et al. (2017), or directly train an OOD detection model to be applied after the initial model Sun et al. (2022a). While effective, they often require tuning and may sacrifice primary task performance.

Distance-based methods treat OOD detection as a separate post-training step Bibas et al. (2021); Hendrycks and Gimpel (2016); Liu et al. (2021); Wang et al. (2022); Sehwag et al. (2021); Sun et al. (2022b); Lee et al. (2018a); Ren et al. (2021); Techapanurak et al. (2019); Chen et al. (2022); Huang et al. (2020); Ming et al. (2023); Denouden et al. (2018); Zhou (2023); Yang et al. (2022); Jiang et al. (2023); Li et al. (2023). They assume OOD data falls far from ID data in the output space and utilize scoring functions like maximum softmax probability Hendrycks and Gimpel (2016), maximum logits Hendrycks et al. (2022a), maximum likelihood Bibas et al. (2021), energy Liu et al. (2021), and reconstruction error Denouden et al. (2018); Zhou (2023); Yang et al. (2022); Jiang et al. (2023); Li et al. (2023) to measure distance between samples. Certain works construct scoring functions in the penultimate layer's feature space Sehwag et al. (2021); Sun et al. (2022b); Chen et al. (2022); Huang et al. (2020); Ming et al. (2023). Distance-based methods are model-agnostic and applicable to pre-trained models if the feature space adequately separates ID and OOD.

We propose a new approach, $HAC_k$-OOD (Hypercone Assisted Contour Generation for OOD Detection), which leverages post-training distance-based OOD concepts. It assumes there is no access to OOD samples, as obtaining a representative sample is infeasible. This allows us to flexibly map

the ID feature space by building a set of multidimensional hypercones, and treating the union of the built hypercones as the new ID class shape.

This paper makes the following contributions. (1) We present, to the best of our knowledge, the first study of class contour generation by employing hypercone projections, effectively providing a new representation for the ID manifold in the feature space. We provide a formalized mathematical definition of the method, and analyze its dynamics in experimental settings. Our work shows the efficacy of this method in the OOD detection setting. (2) Contrary to methods in the literature, we do not make strong distributional assumptions about the feature space, other than that ID and OOD data are separable in the space. The generated contour separates ID from OOD data by considering the variability of the ID data in the directions of the projected hypercones from the class centroid. (3) We show through experimental results that $HAC_k$-OOD+SHE, a variant of $HAC_k$-OOD, reaches state-of-the-art (SOTA) performance in *Far-OOD detection* and *Near-OOD detection* using Supervised Contrastive Learning for CIFAR-100, and performs on par with other SOTA methods on benchmark datasets for CIFAR-10. Experiments with Supervised Cross Entropy show that our method is competitive with SOTA methods on models trained with this loss function. The $HAC_k$-OOD+SHE performance in *Near-OOD detection* detection proves that it performs well even in cases where ID and OOD classes have significant semantic overlap.

## 2 PRELIMINARIES

In line with other distance-based methods for OOD detection Sun et al. (2022b); Sehwag et al. (2021), we frame the task as a multi-class classification problem. We present results on OOD detection for image classification, but this framework can be easily extended beyond image data Liu et al. (2024). Let $X \subseteq \mathbb{R}^D$ represent the input space, where $D = C \times W \times H$, in which $C$ denotes the number of channels, and $W \times H$ denotes the size of the image. The output space $Y$ is defined as $\{1, \ldots, |Y|\}$. The goal of the classification problem is to learn a mapping $f : X \to Y$, which assigns each input observation to one of the $|Y|$ classes. We employ a neural network $f$ trained on samples drawn from the joint distribution $P_{XY}$, where $P_{in}$ represents the marginal distribution over $X$. The network outputs a set of logits, used to predict the label for a given input.

Given a classifier model, such as the one outlined above, our goal during testing is to accurately classify images into one of the $|Y|$ labels (ID), while also being able to detect unknown observations (OOD). Historically, distance-based methods in OOD detection have utilized level set estimation Sun et al. (2022b) in a binary classification approach to determine whether or not observations are drawn from $P_{in}$.

Level set estimation involves partitioning the input space into regions where the classifier's output lies above or below a certain threshold. Let $f(x)$ represent the output (e.g., logits or probabilities) of the classifier for input $x$. The decision boundary is determined by a threshold $\lambda$, such that:

$$\text{Decision}(x) = \mathbf{1}\{S(f(x)) > \lambda\} \tag{1}$$

where $\mathbf{1}\{\cdot\}$ describes the binary classifier in the form of an indicator function. It classifies a sample as ID when the scoring function $S(\cdot)$ produces a score greater than the scalar threshold value $\lambda$. The threshold $\lambda$ is typically chosen based on properties of the training data and/or through validation techniques to optimize performance. This approach effectively creates a boundary in the input space, separating regions where the model is confident in its predictions (ID) from regions where it is uncertain or likely to make errors (OOD).

## 3 RELATED WORK

As mentioned in Section 1, OOD detection methods can be broadly categorized into post-training and training-based approaches. Our proposed method, $HAC_k$-OOD, falls into the post-training category, so we will focus primarily on these techniques while briefly touching on training-based methods for context. In our experiments, we compare $HAC_k$-OOD to several post-training methods, demonstrating its effectiveness across various OOD detection scenarios. Our approach builds upon the strengths of existing feature-space methods, while offering greater flexibility in mapping the embedding space.

### 3.1 Post-Training Methods

Post-training methods can be divided into feature space methods, uncertainty estimation methods, gradient-based methods, activation rectification methods, and hybrid methods, each leveraging different information from the trained model. In this section we introduce key post-training based methods from all categories which will later be used as benchmarks for $HAC_k$-OOD.

**Feature Space Methods:** These methods leverage the rich information encoded in the feature space of neural networks, typically using embeddings from the classifier's penultimate layer. SSD+ Sehwag et al. (2021) assumes a Gaussian distribution of ID observations, while the Mahalanobis method Lee et al. (2018b) uses class-conditional Gaussian distributions for features to define a confidence score based on the Mahalanobis distance. In contrast, KNN+ Sun et al. (2022b) and $HAC_k$-OOD make no distributional assumptions, offering greater flexibility in mapping the embedding space.

**Uncertainty Estimation Methods:** These methods use the final layer outputs. Probability-based approaches like Maximum Softmax Probability (MSP) Hendrycks and Gimpel (2016) and MaxLogit Hendrycks et al. (2022a) classify observations based on maximum softmax probability and maximum logits, respectively. The Generalized Entropy (GEN) method Liu et al. (2023a) introduces an entropy-based score function applicable to any pre-trained softmax-based classifier, designed to amplify minor deviations from ideal one-hot encodings. The Energy method Liu et al. (2021) computes an energy function using logits, attributing higher negative energy values to ID data. Similarly, KL Matching Hendrycks et al. (2022b) forms templates of class posterior distributions, and computes an anomaly score based on the minimum KL divergence between the test input's posterior and these templates.

**Gradient-Based Methods**: GradNorm Huang et al. (2021a) uses gradient space information, noting higher gradient magnitudes for ID data relative to OOD data. It employs the vector norm of gradients, back-propagated from the KL divergence between the softmax output and a uniform probability distribution.

**Activation Rectification Methods**: ReAct and ASH enhance OOD detection by modifying feature activations. ReAct Sun et al. (2021) truncates activations in the classifier's penultimate layer above a specific value (the p-th percentile of model activations) to reduce noise, and aligns activation patterns with well-behaved cases. ASH Djurisic et al. (2023) employs an on-the-fly method to remove a significant portion of a sample's activation at a late layer.

**Hybrid Methods:** ViM Wang et al. (2022) and NNGuide Park et al. (2023) combine information from multiple sources. ViM uses both feature space and logit information, while NNGuide guides classifier-based scores for detection.

### 3.2 Other Methods

**Training-Based Methods:** While not the focus of our work, training-based methods offer complementary approaches to OOD detection. Non-parametric Outlier Synthesis (NPOS) Tao et al. (2023) and Mixture Outlier Exposure (MixOE) Zhang et al. (2021) generate artificial OOD training data. CIDER Ming et al. (2022a), on the other hand, jointly optimizes dispersion and compactness losses to promote ID-OOD separability.

**Multi-Modal Approaches:** Recent work has explored leveraging multiple modalities for OOD detection. Maximum Concept Matching (MCM) Ming et al. (2022b) and LoCoOp Miyai et al. (2024) align visual features with textual concepts utilizing CLIP local features for OOD regularization.

## 4 Method

As embedding-based methods tend to outperform probability-based metrics in distance-based OOD detection, we focus our efforts in this space Ming et al. (2022a); Lu et al. (2024); Sun et al. (2022b); Sehwag et al. (2021). Parametric distance-based methods in the literature, however, necessitate assumptions on the distribution of ID data in the feature space. Thus, they are likely to fail in cases where the distribution has an irregular shape, and are less likely to capture areas of the distribution which do not adhere to the assumptions. $HAC_k$-OOD presents a novel approach to OOD detection which captures the contour of ID data without making assumptions on the distribution of

the embedding space. It uses hypercones which flexibly map the embedding space locally, allowing more precise separation of ID and OOD data.

More specifically, $HAC_k$-OOD extends SSD+ by relaxing the normality assumption that constrains the class contour to a hypersphere (or multidimensional ellipsoid). Instead, $HAC_k$-OOD approximates it with a set of hypercones parameterized by a pre-specified angle. Consequently, our method refrains from assuming a Gaussian distribution for the feature space, and describes its borders not by using multidimensional ellipsoids, but rather by projecting multidimensional hypercones in appropriate directions. This approach allows for a more flexible representation of the class contour.

Unlike the classical distance-based methods, where a single distance cutoff threshold is selected for the full dataset, we adopt a more nuanced strategy. We assign a distinct distance cutoff per projected hypercone, determined by the observed variation in ID distances along that direction. The goal is to accommodate a diverse set of thresholds across different directions. By doing so, our method is unrestricted in shaping the contour of ID observations, fostering greater flexibility and adaptability. Pseudo-code describing hypercone construction and inference is available in Appendix Section A.5.

## 4.1 EMBEDDING EXTRACTION

We use a pre-trained classification network (Section 2), and extract multi-dimensional embedding space features from the penultimate layer which serves as the feature encoder layer. Let $f_{\text{encoder}}(x)$ represent the feature encoder network, a subset of the full classification network, which maps input data $x$ to the extracted features $z$, where $f_{\text{encoder}} : X \rightarrow Z$ is the mapping function from the input space $X$ to the output space $Z$. The training set $X_{\text{train}}$ and the test set $X_{\text{test}}$ for a given supervised classification task are considered to be ID, and we define $X_{\text{ood}}$ to contain the instances of a candidate dataset for the OOD task.

We extract the embedding features of the training set of ID observations $Z_{train} = \{z_{train_1}, \ldots, z_{train_n}\}$, the test set of ID observations $Z_{test} = \{z_{test_1}, \ldots, z_{test_m}\}$, and the unseen test set of observations $Z_{ood} = \{z_{ood_1}, \ldots, z_{ood_v}\}$.

## 4.2 HYPERCONE NOTATION

Let us now introduce key terminology necessary for describing the hypercone. The apex or vertex of a hypercone, $V$, is the central point from which all generating lines originate. The axis of the hypercone, $\vec{a}$, is a straight line passing through the apex, $V$, and some other point $P$. It acts as the central axis of symmetry, defining the primary direction along which the hypercone extends and maintains its symmetry. The slant height of a hypercone is the length of the line segment connecting the apex to any point on the hypercone's surface. The opening angle of the hypercone is the angle between the hypercone axis and any line starting at the apex and extending along its slant height. This opening angle measures how much the hypercone widens or narrows as it extends from the apex along its axis. Mathematically, if we denote a line along the slant height as $\vec{s}$, the opening angle $\theta$ can be expressed as:

$$\cos \theta = \frac{\vec{a} \cdot \vec{s}}{\|\vec{a}\|\|\vec{s}\|} \tag{2}$$

Here, $\cdot$ denotes the dot product and $\| \cdot \|$ denotes the magnitude (length) of a vector. Therefore, for the purposes of this paper, we denote a hypercone $h$ according to its parameters as $(h(\vec{a}, \theta))$. Please refer to the Figure 2 in Appendix Section A.1 for a three dimensional representation of the hypercone and its key components.

## 4.3 HYPERCONE CONSTRUCTION FOR ID DATA CONTOURING

In this section, we describe how to create hypercones, each defined by an axis and opening angle, using the ID features. First, we compute the class contours for the ID training set observations in the embedding space, with one contour per label. The goal is to best describe the boundaries of each class in the embedding space with a set of hypercones. For each class, $HAC_k$-OOD computes its centroid $C_l$ as the mean of all ID train set observations belonging to that class. This creates a set

of centroids, $C$. [1] Each centroid will be the apex of all hypercones for its class. To reposition the embedding features centered at one of the centroids, rather than at the origin of the feature space, $\text{HAC}_k$-OOD computes a centered version of each $Z$.

$$Z_{train} = \{\{z - C_l\} \ \forall \ l \in Y, z \in Z_{train_l}\} \tag{3}$$

$$Z_{test} = \{\{z - C_l\} \ \forall \ l \in Y, z \in Z_{test_l}\} \tag{4}$$

For $Z_{ood}$, we do not have labels. Therefore, we center the embeddings relative to each label, effectively generating a new set of OOD embeddings per label, as follows:

$$Z_{ood} = \{\{z - C_l\} \forall z \in Z_{ood}, l \in Y\} \tag{5}$$

From here onwards, $Z_{train}$, $Z_{test}$ and $Z_{ood}$ will refer to the centered versions of the respective original set of embeddings. $\text{HAC}_k$-OOD now proceeds to construct the set of all hypercones which are parameterized by axis and opening angle. The set of all axes for all hypercones for label $l$ can be defined as:

$$A_l = \{\overrightarrow{C_l z} \ \forall z \in Z_{train_l}\} \tag{6}$$

The set of all axes for all labels can therefore be defined as $A = \{A_l \ \forall l \in Y\}$.

$\text{HAC}_k$-OOD determines each hypercone's opening angle $\theta$ by calculating the cosine distance between its axis and its $k$-th nearest neighbor, where $k$ is a parameter. Given that the axis of the hypercone belongs to one of the train set classes, its set of nearest neighbors is taken to be the set of all train set observations belonging to that class. By determining the angle to the $k$-th nearest neighbor, we ensure that the hypercones include at least $k$ observations within their boundaries. Let KNNAngle($\cdot$) be a function that takes as input a hypercone's axis and the set of all neighbors for the axis, and finds the axis' $k$-th nearest neighbor in cosine distance and consequently the angle between the two. Then, the set of opening angles for all hypercones for label $l$ can be defined as:

$$T_l = \{\text{KNNAngle}(\overrightarrow{\alpha_j}, Z_{train_l}) \ \forall j \in \{1, \ldots, |A_l|\}\} \tag{7}$$

The set of all opening angles for all labels can therefore be defined as $T = \{T_l \ \forall l \in Y\}$.

From Equations 6 and 7, $\text{HAC}_k$-OOD extracts the axes and angles to define the set of hypercones for label $l$. More specifically, for every $j \in \{1, \ldots, |A_l|\}$, it extracts $\theta_j \in T_l$, $\alpha_j \in A_l$, and define $H_l$ as:

$$H_l = \{h(\overrightarrow{a_j}, \theta_j) \ \forall j \in \{1, \ldots, |A_l|\}\} \tag{8}$$

The set of all hypercones for all labels is therefore defined as $H = \{H_l \ \forall l \in Y\}$. The hypercones $H$ initially extend outwards from the pre-computed centroids $C$ without a boundary, serving as filters within the embedding space. While hypercones have a boundary established by a height parameter, we loosely modify this definition to include a radial boundary. To determine the appropriate radial boundary for hypercone $h$, we examine the distribution of $Z_{train_l}$ and $Z_{test_l}$ contained in $h$, or in other words, the ID feature vectors which fall within the angular boundary of hypercone $h$. First, we need to define this set for each $h$. For each $z \in \{Z_{train_l}, Z_{test_l}\}$, we compute the angle between the hypercone axis $\vec{a}$ corresponding to $h$ and the vector $\overrightarrow{C_l z}$ extending from the centroid of the cluster $C_l$ to the feature observation $z$. We denote this angle as $\tau$. If $\tau < \theta$, where $\theta$ is the opening angle of $h$, then $z$ falls within the angular boundary of hypercone $h$.

For a given hypercone $h_{l,i}$, let $G_{l,i}$ be the set of observations falling within its angular boundaries.

---

[1]In cases where the model architecture dictates that the embeddings be normalized, we need to choose different centroids to reflect the new normalized cluster shapes. Normalizing the features effectively projects them onto a unit sphere in the embedding space, resulting in clusters with a disk-like shape. Since the normalized cluster shapes are non-convex, the initial centroids may fall outside the cluster boundaries. To obtain a good approximation of the class centroids within the cluster, each centroid is replaced by its nearest train set observation using cosine distance.

$$G_{l,i} = \{z \ \forall z \in Z_{train_l} \cup Z_{test_l} \mid \tau < \theta_{l,i}\} \tag{9}$$

Such that $\tau$ is the angle between observation $z$, and $\theta_{l,i}$ is the opening angle of $h_{l,i}$. We compute the distances between the apex point $C_l$, of hypercone $h_{l,i}$, and each observation $g \in G_{l,i}$. This set of distances for hypercone $h_{l,i}$ is then given by:

$$D_{l,i} = \{|\overrightarrow{C_l g}| \ \forall g \in G_{l,i}\} \tag{10}$$

We use the distribution of distances in set $D_{l,i}$ to determine a preliminary radial boundary for hypercone $h_{l,i}$, which is taken to be the mean, $\mu$, plus two standard deviations, $2\sigma$, of set $D_{l,i}$.

$$b_{l,i} = \mu + 2\sigma \tag{11}$$

We have chosen this boundary to exclude points which lie far from the class centroid and ensure that the hypercones are robust against outliers. The computed distances are normalized by the radial boundary as follows:

$$D_{l,i}^{norm} = \left\{ \frac{d}{b_{l,i}} \ \forall d \in D_{l,i} \right\} \tag{12}$$

The aforementioned steps are applied to all generated hypercones and observations. This normalization step provides us with the scoring function $S(\cdot)$ for $\text{HAC}_k$-OOD as defined in 1. The computed scores can then be used in level set estimation (from Section 2), and ensure that the results are reported at a pre-determined true positive rate (TPR). The TPR is set to 95%, effectively ensuring that 95% of all ID observations are correctly classified as in distribution. The score at the 95-th percentile, $\lambda$, effectively becomes the final radial boundary of the hypercones. The final contour per class comprises of the union of the constructed hypercones for that class. Please refer to Figure 3 in Appendix Section A.2 which illustrates the process of hypercone construction.

## 4.4 OOD INFERENCE

During inference, the hypercones are employed to determine whether a new observation in the embedding space, $z$, is ID or OOD. This decision is made by checking whether or not the observation falls within both the angular and radial boundaries of any of the generated hypercones, in any of the clusters, using the same method described in Section 4.3. We use $z \in h_i$ to mean that observation $z$ falls within both angular and radial boundaries of hypercone $h_i$. If it does, it is labeled as ID, and OOD otherwise. Thus, the level-set estimation formulation from Section 2 transforms to an OOD detector framework defined as:

$$\text{Decision}(z) = \begin{cases} \text{ID} & \text{if } \exists \ h_i \in H_l \ \forall l \in Y \ \text{s.t. } z \in h_i \\ \text{OOD} & \text{otherwise} \end{cases} \tag{13}$$

The hypercones inherently aim to delineate the contour of ID observations by allowing for fluid boundaries between ID and OOD observations in different areas of the embedding space, as opposed to existing approaches that rely on a single distance threshold for the entire space Sehwag et al. (2021); Sun et al. (2022b). Additionally, by utilizing ID observations as the hypercone axes, we not only ensure that we generate the contour by scanning the appropriate directions, but also facilitate the generation of overlapping hypercones in densely populated areas of the embedding space. This approach smooths out the contour's surface, dimming the effects of outliers, akin to fitting a polynomial curve using interpolation techniques.

## 5 EXPERIMENTS

### 5.1 BENCHMARKS AND EVALUATION METRICS

We evaluate $\text{HAC}_k$-OOD relative to 12 other post-training OOD detection methods: MSP Hendrycks and Gimpel (2016), Mahalanobis Lee et al. (2018c) MaxLogit Hendrycks et al. (2022a), Energy

Liu et al. (2021), ViM Wang et al. (2022), GradNorm Huang et al. (2021b), SSD+ Sehwag et al. (2021), KL matching Hendrycks et al. (2019b), KNN+ Sun et al. (2022b), GEN Liu et al. (2023b), NNGuide Park et al. (2023), and SHE Zhang et al. (2023). We also combine $\text{HAC}_k$-OOD and Mahalanobis Lee et al. (2018c) with Simplified Hopfield Energy (SHE) Zhang et al. (2023). SHE introduces a "store-then-compare" framework, transforming penultimate layer outputs into stored patterns representing ID data, which we have used in a potentially novel way as centroids in $\text{HAC}_k$-OOD+SHE and Mahalanobis+SHE. For consistency, we reproduce the results of the 12 benchmarks as well as the additional variants of $\text{HAC}_k$-OOD and Mahalanobis. Experiments combining $\text{HAC}_k$-OOD and activation clipping method ReAct are found in Appendix Section A.4. The metrics we report on are consistent with standard metrics in the OOD literature: the false positive rate of OOD data when the TPR is 95% (FPR95), and AUROC.

## 5.2 CLASSIFICATION NETWORKS

We train two classification networks. The first one is a ResNet trained on ID data using NT-Xent Sohn (2016) for Supervised Contrastive Learning with an embedding dimension of 128, a batch size of 2000, learning rate of 0.5, and cosine annealing for 500 epochs. The network is warmed up for 10 epochs. For logit-based methods, we train a linear classifier on top of the trained backbone as in Khosla et al. (2020). Moreover, we extract the logits from the last layer of the network. For embedding-based methods, we extract the embeddings from the penultimate layer of the network. The second classification network is identical to the first one, but using a cross-entropy loss to show that $\text{HAC}_k$-OOD is training agnostic. We expect to obtain better results using Supervised Contrastive Learning, as it is known to generate embeddings with a greater degree of separability.

## 5.3 DATASETS

We test $\text{HAC}_k$-OOD's performance on CIFAR-100 Krizhevsky (2009) as our ID dataset. It has 100 classes, and is considered a challenging dataset in the OOD detection literature. In Appendix Section A.4, we present results for CIFAR-10 (see tables 4, 5), which represents a simpler case with only 10 classes. We evaluate $\text{HAC}_k$-OOD's performance for five OOD datasets: Textures Cimpoi et al. (2014), iSUN Xu et al. (2015), LSUN Yu et al. (2016), Places365 Zhou et al. (2018), and SVHN Netzer et al. (2011). We also evaluate its performance for CIFAR-100 on *Near-OOD detection* on special LSUN and Imagenet Deng et al. (2009) splits proposed by Tack et al. (2020) along with CIFAR-10 an OOD dataset this time.

## 5.4 $\text{HAC}_k$-OOD PARAMETERS

As discussed in Section 4, each hypercone's opening angle is determined by the cosine distance to its $k$-th nearest neighbor, where $k$ is a tunable parameter. We propose an automatic selection method called *Adaptive $k$*, which, though not optimal, performs well across datasets and architectures. Selecting $k$ optimally would require a holdout OOD set, but, as noted in Section 1, we assume no access to such data. Thus, we take a heuristic approach that chooses a specific $k$ for each $l \in Y$ by regularizing an informed upper bound for $k$ by a factor of the number of class observations and feature dimensions, while at the same time incorporating the point density of the class. This is further discussed in Appendix Section A.3. The objective of using this method to choose $k$ is to remove the burden of searching for the best $k$, which is further explored in 5.6.

## 5.5 RESULTS

Table 1 shows the results for CIFAR-100 trained on ResNet-18, 34, and 50 with Supervised Contrastive Learning. The results indicate that using SHE's stored patterns as centroids significantly enhances performance compared to using the original centroids, producing state-of-the art performance in both average FPR95 and AUROC on all tested architectures. Moreover, $\text{HAC}_k$-OOD+SHE shows a clear performance increase as the size of the classification network grows, with FPR95 values of 51.86%, 46.93%, and 35.91% for ResNet-18, 34, and 50 respectively, resulting in a 10.03% and 2.47% gap in average FPR95 and AUROC respectively over the best baseline method when using ResNet-50. We attribute this effect to two main factors: (1) Larger networks create better separation between classes, which $\text{HAC}_k$-OOD+SHE and $\text{HAC}_k$-OOD can leverage and (2) SHE improves

centroid calculation by excluding noisy training samples, using only correctly classified samples, allowing HACk-OOD+SHE to more accurately position the centroid, reduce angular errors, and prevent the over-extension of radial boundaries.

Elaborating upon the first factor, unlike most baseline methods, $HAC_k$-OOD creates class-specific decision boundaries rather than a single global boundary for all classes. Larger networks can better capture subtle patterns in the data, such as sub-hierarchies within clusters. Distribution assumption-free methods, like $HAC_k$-OOD, SHE, NNGuide and KNN+, are better suited to handle these cases as they do not assume normality and can vary in different directions. In contrast, smaller models like ResNet-18 struggle to capture these subtle patterns, making baseline methods with simplifying assumptions, such as SSD+ and Mahalanobis, perform better.

The results in Table 2 demonstrate significant improvements in *Near-OOD detection* performance over previous SOTA methods on CIFAR-100. *Near-OOD detection* is a more challenging task due to the similarity between unseen observations and the ID dataset. As in the *Far-OOD detection* experiments, $HAC_k$-OOD+SHE consistently improves as the capacity of the network increases (78.46%, 74.61%, and 73.89% FPR95 on ResNet-18, 34, and 50 respectively), leading to SOTA performance in both average FPR95 and AUROC on ResNet-34 and ResNet-50. Moreover, ResNet-50 not only surpasses other architectures but also widens the performance gap between $HAC_k$-OOD+SHE and the best baseline method, from 2.12% to 6.81% in average FPR95 and from 0.53% to 2.27% in AUROC. Additionally, $HAC_k$-OOD+SHE and $HAC_k$-OOD outperform all baseline methods in three out of five datasets across both metrics. This supports our theory that $HAC_k$-OOD excels due to its ability to generate class contours that better capture the dataset's variability, which is crucial for *Near-OOD detection*. Consistent with the results in Table 1, $HAC_k$-OOD's performance improves with larger classifiers, enhancing cluster separability. ResNet-18 results further confirm our previous conclusions.

Experiments show that $HAC_k$-OOD+SHE is computationally efficient, having an average inference time of 1.00, 0.95 and 2.22 ms per sample on ResNet-18, 34 and 50 respectively on an 16 core, 128GB RAM server.

Finally, Table 3, in Appendix Section A.4, shows a similar trend for $HAC_k$-OOD and $HAC_k$-OOD+SHE when models are trained with Cross Entropy Loss. $HAC_k$-OOD and $HAC_k$-OOD+SHE show consistent performance improvement with increased network capacity, resulting in a 16.09% and 16.20% drop in FPR95 respectively from ResNet-18 to ResNet-50. Furthermore, $HAC_k$-OOD+SHE outperforms baseline methods in 2 out of 5 OOD datasets on ResNet-34 and ResNet-50, achieving SOTA performance in both average FPR95 and AUROC on ResNet-34. These results align with the Supervised Contrastive Learning results in Tables 1 and 2, further supporting our hypothesis.

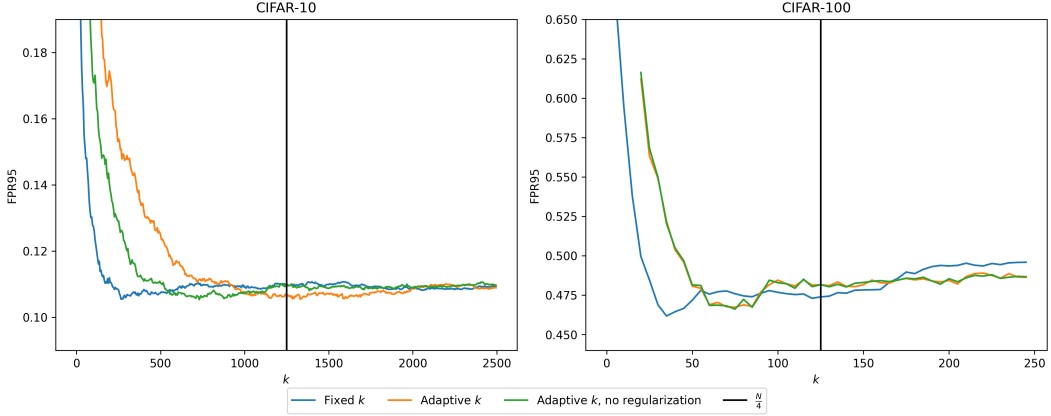

Figure 1: Relationship between hyperparameter $k$ and FPR95 on CIFAR-10 (left) and CIFAR-100 (right) for ResNet-34 Supervised Contrastive classifier features. The blue line shows $HAC_k$-OOD for fixed $k$ values. The orange line represents *Adaptive $k$* with different $k$ values per label and regularization. The green line shows *Adaptive $k$* without regularization.

Table 1: *Far-OOD detection* CIFAR-100 Supervised Contrastive Learning.

| Backbone | Method | Textures | | iSUN | | LSUN | | Places365 | | SVHN | | Average | |
|---|---|---|---|---|---|---|---|---|---|---|---|---|---|
| | | FPR95 ↓ | AUROC ↑ | FPR95 ↓ | AUROC ↑ | FPR95 ↓ | AUROC ↑ | FPR95 ↓ | AUROC ↑ | FPR95 ↓ | AUROC ↑ | FPR95 ↓ | AUROC ↑ |
| ResNet-18 | MSP | 78.55 | 79.83 | 72.92 | 82.29 | 75.62 | 79.85 | 80.51 | 77.34 | 70.18 | 84.77 | 75.56 | 80.82 |
| | MaxLogit | 74.29 | 83.92 | 67.5 | 86.03 | 68.81 | 85.04 | 79.02 | 78.95 | 65.54 | 87.73 | 71.03 | 84.33 |
| | Energy | 68.56 | 85.00 | 62.54 | 87.04 | 60.87 | 86.6 | 78.02 | 79.13 | 60.62 | 88.49 | 66.12 | 85.25 |
| | ViM | 62.43 | 86.92 | 86.15 | 82.44 | 50.14 | 90.63 | 78.12 | 79.09 | 27.59 | 94.8 | 60.89 | 86.78 |
| | GradNorm | 71.81 | 83.31 | 64.53 | 85.82 | 64.57 | 84.64 | 78.71 | 78.77 | 62.52 | 87.72 | 68.43 | 84.05 |
| | Mahalanobis | 55.05 | 88.79 | 85.55 | 82.42 | 40.36 | 92.81 | 76.32 | 79.84 | 19.3 | 96.32 | 55.32 | 88.04 |
| | KL Matching | 75.83 | 79.78 | 69.7 | 82.43 | 74.78 | 79.23 | 78.22 | 77.07 | 69.66 | 84.53 | 73.64 | 80.61 |
| | KNN+ | 55.9 | 87.91 | 74.06 | 84.92 | 48.72 | 89.14 | 79.36 | 77.35 | 44.49 | 91.84 | 60.51 | 86.23 |
| | SSD+ | 52.68 | 89.51 | 89.78 | 78.56 | 28.84 | **95.07** | 80.49 | 76.88 | **18.76** | **96.44** | 54.11 | 87.29 |
| | GEN | 69.18 | 84.78 | 63.2 | 86.87 | 61.7 | 86.32 | 78.29 | 79.09 | 61.14 | 88.38 | 66.70 | 85.09 |
| | NNGuide | 64.2 | 86 | 67.73 | 85.9 | 59.37 | 86.81 | 78.53 | 78.87 | 59.61 | 88.95 | 65.89 | 85.31 |
| | SHE | 52.48 | 88.89 | 76.08 | 83.03 | 52.14 | 88.24 | 81.93 | 75.5 | 57.2 | 89.57 | 63.97 | 85.05 |
| | ASH | 41.37 | 91.38 | 70.76 | 83.62 | 27.61 | 95.02 | 78.31 | 78.15 | 63.02 | 87.05 | 56.21 | 87.04 |
| | SCALE | 37.68 | 92.03 | 71.8 | 82.62 | 25.02 | 95.53 | 78.21 | 77.82 | 63.33 | 87.00 | 55.21 | 87.00 |
| | HAC$_k$-OOD | 52.7 | 88.03 | 75.91 | 83.52 | 31.89 | 93.52 | 76.30 | 79.71 | 52.95 | 89.78 | 57.95 | 86.91 |
| | Mahalanobis+SHE | 55.04 | 88.79 | 85.58 | 82.43 | 40.38 | 92.81 | 76.34 | 79.84 | 19.30 | 96.32 | 55.33 | 88.04 |
| | HAC$_k$-OOD+SHE | **45.89** | **90.23** | 73.38 | 84.9 | **25.16** | **95.07** | **72.97** | **80.38** | 41.88 | 92.06 | **51.86** | **88.53** |
| ResNet-34 | MSP | 74.13 | 82.44 | 72.67 | 83.19 | 75.34 | 81.66 | 80.00 | 77.97 | 65.36 | 86.76 | 73.50 | 82.40 |
| | MaxLogit | 69.18 | 85.26 | 68.55 | 85.51 | 70.69 | 84.54 | 79.14 | 79.06 | 62.33 | 88.50 | 69.98 | 84.57 |
| | Energy | 63.94 | 86.03 | 65.12 | 86.08 | 66.21 | 85.28 | 78.03 | 79.23 | 59.42 | 88.89 | 66.54 | 85.10 |
| | ViM | 51.28 | 89.61 | 76.85 | 85.00 | 49.35 | 91.69 | 76.64 | 80.14 | 25.14 | 95.28 | 55.85 | 88.34 |
| | GradNorm | 65.76 | 84.96 | 65.29 | 85.52 | 68.00 | 84.26 | 78.14 | 79.10 | 59.23 | 88.70 | 67.28 | 84.51 |
| | Mahalanobis | 47.75 | 90.22 | 72.75 | 85.59 | 38.35 | 93.14 | 74.80 | 80.79 | 19.69 | 96.28 | 50.67 | 89.20 |
| | KL Matching | 76.45 | 81.47 | 71.07 | 82.78 | 76.25 | 80.91 | 79.14 | 76.94 | 66.06 | 86.35 | 73.79 | 81.69 |
| | KNN+ | 53.65 | 88.43 | 66.03 | 86.39 | 49.59 | 90.61 | 76.52 | 79.85 | 36.72 | 93.43 | 56.5 | 87.74 |
| | SSD+ | **42.27** | **91.78** | 74.76 | 85.09 | 29.57 | 94.82 | 76.1 | 80.17 | **17.87** | **96.69** | 48.11 | 89.71 |
| | GEN | 64.15 | 85.89 | 65.10 | 86.01 | 66.62 | 85.15 | 77.96 | 79.21 | 59.31 | 88.86 | 66.63 | 85.02 |
| | NNGuide | 58.58 | 87.36 | 65.68 | 86.02 | 58.97 | 87.63 | 76.88 | 79.67 | 48.54 | 90.89 | 61.73 | 86.31 |
| | SHE | 52.02 | 89.14 | 67.71 | 85.74 | 51.71 | 90.12 | 77.85 | 79.22 | 39.7 | 92.99 | 57.8 | 87.44 |
| | ASH | 39.79 | 91.09 | 64.90 | 83.40 | 36.34 | 93.52 | 77.86 | 77.42 | 67.99 | 82.09 | 57.38 | 85.50 |
| | SCALE | 34.33 | 90.38 | 67.88 | 78.66 | 27.66 | 94.11 | 78.86 | 73.12 | 71.54 | 75.23 | 56.05 | 82.30 |
| | HAC$_k$-OOD | 46.17 | 90.08 | **61.14** | **87.47** | 33.17 | 93.61 | 71.31 | 81.29 | 29.01 | 94.47 | 48.16 | 89.38 |
| | Mahalanobis+SHE | 47.71 | 90.22 | 72.75 | 85.59 | 38.34 | 93.13 | 74.80 | 80.79 | 19.68 | 96.28 | 50.66 | 89.20 |
| | HAC$_k$-OOD+SHE | 44.82 | 90.34 | 62.3 | 87.15 | **28.71** | **94.48** | **70.39** | **81.6** | 28.43 | 94.54 | **46.93** | **89.62** |
| ResNet-50 | MSP | 73.90 | 84.52 | 81.30 | 78.08 | 76.70 | 85.07 | 79.96 | 79.07 | 60.69 | 89.01 | 74.51 | 83.15 |
| | MaxLogit | 69.93 | 86.99 | 79.34 | 80.72 | 72.16 | 87.80 | 78.89 | 79.83 | 55.9 | 90.64 | 71.24 | 85.20 |
| | Energy | 64.13 | 87.96 | 76.36 | 81.43 | 65.58 | 89.02 | 77.92 | 79.97 | 51.29 | 91.26 | 67.06 | 85.93 |
| | ViM | 63.44 | 86.52 | 95.87 | 74.29 | 71.71 | 87.48 | 78.17 | 79.95 | 8.37 | 98.4 | 63.51 | 85.33 |
| | GradNorm | 67.38 | 87.02 | 78.07 | 80.37 | 68.66 | 88.22 | 78.74 | 79.85 | 52.53 | 90.92 | 69.08 | 85.28 |
| | Mahalanobis | 50.04 | 89.96 | 95.60 | 74.06 | 51.44 | 92.46 | 77.12 | 80.37 | 5.75 | 98.90 | 55.99 | 87.15 |
| | KL Matching | 89.93 | 81.26 | 83.85 | 77.03 | 88.74 | 82.34 | 81.51 | 78.12 | 68.04 | 87.67 | 82.41 | 81.28 |
| | KNN+ | 34.26 | 93.14 | 78.07 | 80.84 | 30.28 | 94.71 | 75.39 | 80.56 | 13.46 | 97.68 | 46.29 | 89.39 |
| | SSD+ | 48.01 | 90.28 | 96.03 | 72.46 | 52.3 | 92.37 | 78.84 | 79.39 | **5.24** | **99.01** | 56.08 | 86.70 |
| | GEN | 64.82 | 87.80 | 76.77 | 81.3 | 66.46 | 88.86 | 78.04 | 79.96 | 51.82 | 91.19 | 67.58 | 85.82 |
| | NNGuide | 48.95 | 90.5 | 79.82 | 80.3 | 50.43 | 91.55 | 76.53 | 80.45 | 34.55 | 94.05 | 58.06 | 87.37 |
| | SHE | 26.74 | 94.67 | 79 | 80.86 | 29.62 | 94.81 | 77.03 | 80.08 | 17.32 | 96.99 | 45.94 | 89.48 |
| | ASH | 19.11 | 95.81 | 67.46 | 85.4 | 16.27 | 97.09 | 78.54 | 78.03 | 31.74 | 94.22 | 42.62 | 90.11 |
| | SCALE | 18.63 | 95.94 | 64.83 | 87.13 | 15.95 | 97.21 | 76.68 | 79.41 | 32.76 | 94.31 | 41.77 | 90.80 |
| | HAC$_k$-OOD | 34.29 | 92.95 | 63.13 | **86.72** | 15.86 | 97.12 | 71.93 | 82.12 | 14.66 | 97.09 | 39.97 | 91.20 |
| | Mahalanobis+SHE | 50.02 | 89.97 | 95.60 | 74.05 | 51.42 | 92.46 | 77.11 | 80.36 | 5.74 | 98.90 | 55.98 | 87.15 |
| | HAC$_k$-OOD+SHE | **26.49** | **94.59** | **62.81** | 86.15 | **11.68** | **97.84** | **69.07** | **83.00** | 9.51 | 98.19 | **35.91** | **91.95** |

## 5.6 ABLATIONS

We now present ablation studies on choosing the value of $k$ for HAC$_k$-OOD and on the effect that hypercone axes directions have on HAC$_k$-OOD's performance.

### 5.6.1 PARAMETERS

HAC$_k$-OOD only relies on one key parameter: the number of nearest neighbors ($k$) used to compute the hypercone opening angle. As the opening angle increases, the number of observations within each hypercone grows. A smaller angle provides a more precise contour of ID observations, assuming sufficient data-points to avoid gaps between hypercones where ID observations may go undetected. Narrow hypercones may also fail to represent low-density areas accurately. Conversely, a larger angle captures more ID observations but risks including OOD observations. Figure 1 shows FPR95 for different values of $k$. The blue line represents FPR95 with a fixed $k$ for all labels, while the orange and green lines show FPR95 using *Adaptive* $k$. *Adaptive* $k$ involves an additional regularization factor, so we test the effect with and without regularization, respectively. In the latter two cases, $k$ represents the maximum value of $k$ so the actual value for each class may vary, but will always be less than or equal to this value. The vertical black line marks $\frac{N}{4}$, the maximum $k$ used in our main experiments. Its intersection with the orange line represents the results reported in Section 5.5.

The FPR95 decreases sharply as $k$ increases, reaching a minimum before gradually rising again. As long as $k$ is not too small, its impact on the results remains limited. However, *Adaptive* $k$ chooses a value of $k$ for each label that yields an FPR95 close to the observed minimum, making it a strong approximation of the optimal $k$. We find that the regularization factor plays a crucial role in guiding *Adaptive* $k$ to select an effective $k$, particularly in datasets like CIFAR-10, where class sizes are large.

Table 2: *Near-OOD detection* CIFAR-100 Supervised Contrastive Learning.

| Backbone | Method | OOD Datasets | | | | | | | | | |
|---|---|---|---|---|---|---|---|---|---|---|---|
| | | LSUN-F | | Imagenet-F | | Imagenet-R | | CIFAR-10 | | Average | |
| | | FPR95 ↓ | AUROC ↑ | FPR95 ↓ | AUROC ↑ | FPR95 ↓ | AUROC ↑ | FPR95 ↓ | AUROC ↑ | FPR95 ↓ | AUROC ↑ |
| | MSP | 85.35 | 74.24 | 79.36 | 77.43 | 73.68 | 81.63 | 82.9 | 76.02 | 80.32 | 77.33 |
| | MaxLogit | 86.26 | 74.95 | 74.14 | 78.94 | 69.59 | 85.13 | 83.29 | 76.24 | 79.32 | 78.82 |
| | Energy | 86.64 | 74.78 | 76.49 | 79.14 | 65.41 | 86.01 | 83.12 | 76.09 | 77.91 | 79.00 |
| | ViM | 85.91 | 74.49 | 76.55 | 79.09 | 84.26 | 81.58 | 84.58 | 74.01 | 82.82 | 77.29 |
| | GradNorm | 87.11 | 74.68 | 76.89 | 78.96 | 67.31 | 84.94 | 82.81 | 76.58 | 78.53 | 78.79 |
| | Mahalanobis | 84.73 | 75.28 | 76.54 | 79.34 | 84.26 | 81.04 | 85.43 | 73.14 | 82.74 | 77.20 |
| | KL-Matching | 83.17 | 73.71 | 78.14 | 77.08 | 71.58 | 81.63 | 82.51 | 75.13 | 78.85 | 76.89 |
| ResNet-18 | KNN+ | 89.51 | 72.43 | 77.59 | 78.62 | 73.98 | 84.36 | 84.97 | 72.65 | 81.51 | 77.02 |
| | SSD+ | 87.93 | 71.57 | 78.86 | 76.87 | 87.05 | 77.47 | 87.69 | 68.32 | 85.38 | 73.56 |
| | GEN | 86.78 | 74.79 | 76.6 | 79.13 | 66.01 | 85.87 | 83.19 | 76.15 | 78.14 | 78.99 |
| | NNGuide | 87.42 | 74.51 | 76.64 | 79.1 | 69.25 | 84.83 | 83.72 | 75.03 | 79.26 | 78.37 |
| | SHE | 91.09 | 70.70 | 80.08 | 77.04 | 75.31 | 83.13 | 85.50 | 71.45 | 83.00 | 75.58 |
| | ASH | 87.05 | 71.50 | 79.76 | 75.69 | 71.31 | 81.86 | 86.86 | 67.91 | 81.24 | 74.24 |
| | SCALE | 88.59 | 70.2 | 80.5 | 74.4 | 73.12 | 80.19 | 88.26 | 64.85 | 82.62 | 72.41 |
| | HAC$_k$-OOD | 76.62 | 78.99 | 77.38 | 78.52 | 79.27 | 81.25 | 87.2 | 71.74 | 80.12 | 77.62 |
| | Mahalanobis+SHE | 84.75 | 75.28 | 76.56 | 79.34 | 84.28 | 81.04 | 85.42 | 73.14 | 82.75 | 77.20 |
| | HAC$_k$-OOD+SHE | 78.64 | 77.33 | 74.98 | 79.25 | 76.52 | 82.96 | 83.72 | 74.01 | 78.46 | 78.39 |
| | MSP | 85.59 | 74.55 | 77.99 | 78.29 | 72.89 | 82.71 | 82.35 | 77.15 | 79.7 | 78.18 |
| | MaxLogit | 85.56 | 74.98 | 76.62 | 79.34 | 69.8 | 84.85 | 82.18 | 77.28 | 78.54 | 79.11 |
| | Energy | 85.62 | 74.95 | 75.13 | 79.51 | 67.08 | 85.33 | 82.28 | 77.22 | 77.53 | 79.25 |
| | ViM | 82.51 | 77.34 | 75.08 | 80.23 | 75.5 | 84.7 | 84.18 | 75.73 | 79.32 | 79.5 |
| | GradNorm | 85.82 | 75.12 | 75.02 | 79.39 | 67.12 | 84.91 | 81.99 | 77.67 | 77.49 | 79.27 |
| | Mahalanobis | 80.83 | 78.20 | 73.29 | 80.93 | 73.26 | 85.03 | 83.93 | 76.09 | 77.83 | 80.06 |
| | KL-Matching | 83.69 | 73.62 | 77.42 | 77.45 | 71.73 | 82.3 | 81.54 | 75.93 | 78.6 | 77.32 |
| ResNet-34 | KNN+ | 84.04 | 77.22 | 73.5 | 80.75 | 66.15 | 86.02 | 83.23 | 76.51 | 76.73 | 80.12 |
| | SSD+ | 81.51 | 77.9 | 75.03 | 80.19 | 73.48 | 84.57 | 85.76 | 73.55 | 78.95 | 79.05 |
| | GEN | 85.59 | 74.96 | 75.12 | 79.49 | 66.97 | 85.28 | 82.08 | 77.27 | 77.44 | 79.25 |
| | NNGuide | 84.74 | 75.67 | 73.76 | 79.81 | 66.66 | 85.32 | 82.61 | 77.01 | 76.94 | 79.45 |
| | SHE | 84.42 | 76.73 | 74.48 | 80.23 | 67.39 | 85.34 | 84.08 | 75.49 | 77.59 | 79.45 |
| | ASH | 84.56 | 72.68 | 78.17 | 74.64 | 68.45 | 80.02 | 87.00 | 65.36 | 79.54 | 73.18 |
| | SCALE | 86.57 | 68.32 | 81.18 | 68.08 | 71.78 | 73.07 | 90.61 | 55.03 | 82.54 | 66.12 |
| | HAC$_k$-OOD | 74.59 | 78.91 | 73.8 | 80.35 | 68.6 | 85.64 | 84.03 | 76.68 | 75.26 | 80.4 |
| | Mahalanobis+SHE | 80.81 | 78.20 | 73.28 | 80.93 | 73.24 | 85.02 | 83.92 | 76.09 | 77.81 | 80.06 |
| | HAC$_k$-OOD+SHE | 73.74 | 79.42 | 72.67 | 80.75 | 68.79 | 85.41 | 83.23 | 77.03 | 74.61 | 80.65 |
| | MSP | 85.87 | 75.46 | 76.77 | 79.65 | 84.85 | 75.91 | 81.04 | 78.08 | 82.13 | 77.28 |
| | MaxLogit | 86.48 | 75.56 | 74.84 | 80.54 | 83.17 | 78.14 | 81.07 | 77.93 | 81.39 | 78.04 |
| | Energy | 86.43 | 75.36 | 73.69 | 80.75 | 81.33 | 78.68 | 81.34 | 77.8 | 80.7 | 78.15 |
| | ViM | 84.91 | 76.03 | 76.16 | 80.21 | 95.54 | 70.86 | 87.31 | 75.17 | 85.98 | 75.57 |
| | GradNorm | 87.17 | 75.46 | 73.69 | 80.61 | 82.44 | 77.83 | 80.85 | 78.28 | 81.04 | 78.04 |
| | Mahalanobis | 83.23 | 76.98 | 75.87 | 80.00 | 95.16 | 70.99 | 89.19 | 73.04 | 85.86 | 75.25 |
| | KL Matching | 83.17 | 74.85 | 78.7 | 78.71 | 86.57 | 75.01 | 78.94 | 77.39 | 81.84 | 76.49 |
| ResNet-50 | KNN+ | 85.94 | 76.31 | 73.11 | 81.02 | 80.47 | 77.86 | 87.53 | 74.34 | 81.76 | 77.38 |
| | SSD+ | 85.1 | 75.78 | 77.91 | 78.8 | 95.43 | 69 | 91.02 | 70.13 | 87.36 | 73.43 |
| | GEN | 86.62 | 75.39 | 73.82 | 80.73 | 81.69 | 78.57 | 81.29 | 77.86 | 80.86 | 78.14 |
| | NNGuide | 86.47 | 75.87 | 71.93 | 81.06 | 82.97 | 77.39 | 83.07 | 76.67 | 81.11 | 77.75 |
| | SHE | 87.15 | 76.22 | 74.75 | 80.29 | 79.91 | 78.63 | 88.05 | 72.5 | 82.46 | 76.91 |
| | ASH | 86.77 | 72.52 | 77.35 | 75.74 | 71.11 | 83.29 | 89.24 | 62.97 | 81.12 | 73.63 |
| | SCALE | 87.29 | 73.01 | 78.11 | 76.15 | 71.11 | 84.5 | 90.10 | 61.45 | 81.65 | 73.78 |
| | HAC$_k$-OOD | 72.65 | 80.38 | 69.90 | 81.73 | 69.56 | 84.18 | 84.27 | 76.11 | 74.10 | 80.6 |
| | Mahalanobis+SHE | 83.23 | 76.98 | 75.83 | 80.00 | 95.16 | 70.99 | 89.15 | 73.04 | 85.84 | 75.25 |
| | HAC$_k$-OOD+SHE | 72.99 | 80.07 | 68.00 | 82.19 | 69.86 | 83.32 | 84.71 | 76.07 | 73.89 | 80.41 |

### 5.6.2 HYPERCONE AXES DIRECTIONS

Aligning the hypercone axes with ID train set observations is a highly effective technique for accurately approximating the contour. This approach correctly identifies the majority of ID observations, while efficiently filtering out most OOD instances. However, randomizing these directions proves ineffective. Particularly, using uniformly sampled hypercone axis directions increases the FPR95 by 14.36% for ResNet-34 trained on CIFAR-100 and by 64.42% when trained on CIFAR-10.

## 6 CONCLUSION

This paper introduces HAC$_k$-OOD, a novel approach to post-training OOD detection. It constructs class contours in a classifier's embedding space using multi-dimensional hypercone projections. Our method demonstrates SOTA performance in challenging feature spaces, and performs comparably to other SOTA methods in simpler feature spaces. We plan to address in the future the optimal selection of both $k$ and the preliminary radial boundary, as well as explore the effect of different centroids on HAC$_k$-OOD's performance. We look forward to exploring the potential of hypercone-assisted contour generation for other applications, such as classification and feature space explainability.

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

# A APPENDIX

## A.1 HYPERCONE IN 3D

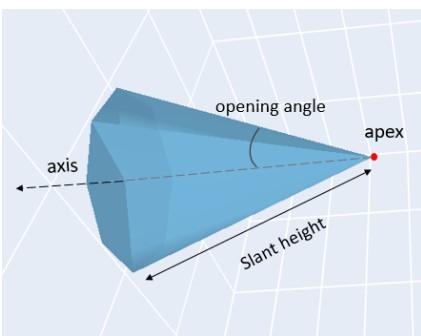

Figure 2: Hypercone in 3D space, showing the axis, opening angle, apex, and slant height.

## A.2 CONTOUR CONSTRUCTION IN 2D

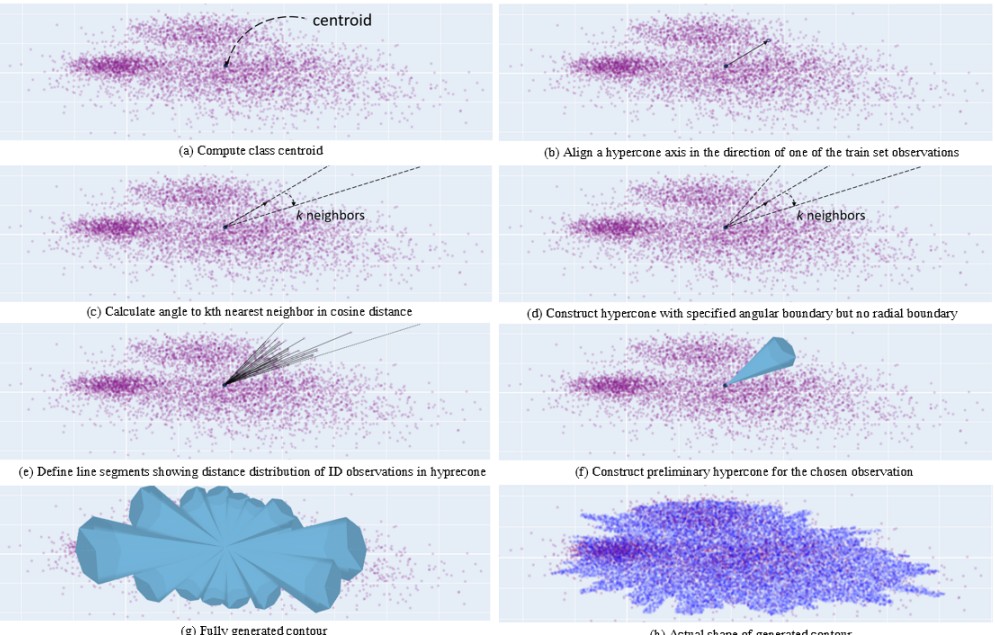

Figure 3: HAC$_k$-OOD steps to generate a class contour in two dimensions. It illustrates what a single ID cluster's contour would look like in two dimensional space. Data was generated by drawing 5000 observations sampled from 5 Gaussian distributions and placing the cluster means sufficiently close to represent one larger cluster which varies non-uniformly. Sub-figures (a)-(f) show a representation of how one hypercone is constructed, sub-figure (g) shows a model representation of what the shape of the expected contour would be. Sub-figure (h) shows in blue the actual shape of HAC$_k$-OOD's contour when running HAC$_k$-OOD on the this synthetic data.

## A.3 HEURISTIC APPROACH FOR CHOOSING $k$

We use $\theta$ as a proxy to arrive at the maximum value for $k$. Here, we would like to remind the reader that the opening angle of the hypercone is only half of the total angle the hypercone spans (see Figure 2). We take the limiting case of a uniformly distributed two dimensional class. To maintain the

convexity of the hypercones, $\frac{\pi}{2}$ is chosen as the maximum allowable angle. As $N \to \infty$, selecting $k = \frac{N}{4}$ neighbors yields this angle, where $N$ is the number of observations in a class.

In higher dimensions, however, $\frac{\pi}{2}$ covers a much smaller portion of the space, so $k = \frac{N}{4}$ serves as a conservative upper bound for $k$ in a space with greater than two dimensions.

In making a final choice for $k$ per class we:

1. Scale $k$ using the inverse logarithmic function of the ratio between the number of observations in the class and the dimensionality. The distribution of $n$ points in the $d$ dimensional space should affect the choice of $k$. A large number of points in a small dimensional space allows for narrower hypercones to be generated, while fewer points in a higher dimensional space necessitates broader hypercones. In order to accommodate this distributional effect on $k$, we scale it using a regularization factor $\zeta$ as follows:

$$k = k * \zeta(n, d) \tag{14}$$

where,

$$\zeta(n, d) = \frac{1}{1 + \log(n/d)} \tag{15}$$

2. Additionally, given that $k = \frac{N}{4}$ depends on a uniformly distributed space, we scale the value by the ratio of point density between our class and a uniformly distributed class of approximately the same size. We generate a synthetic dataset of features $z_i$ drawn from $U(\alpha, \beta)$ such that $Z_{uniform} = \{z_i \in \mathbb{R}^n \; \forall i \in \{1, \ldots, N\} | z_i \sim U(\alpha, \beta)\}$, where $\alpha$ is the minimum value of our class feature observations and $\beta$ the maximum, with the same dimensions as our class. We compute the cosine distance to the $i$-th nearest neighbor, where $i \in \{f * k \; \forall f \in \{0.05, 0.10, 0.15, \ldots, 1\} \mid k = \frac{N}{4}\}$, for both the original class in our dataset and the synthetic uniformly distributed class. The ratio of mean cosine distance values between the two classes is also used to scale the value $k$.

Note that the final approximation for $k$ is not optimal and in future work we plan to explore optimal selection of $k$.

### A.4 MORE EXPERIMENTS

Results on Table 3 show that although we can reach SOTA in ResNet-34, $\text{HAC}_k$-OOD and $\text{HAC}_k$-OOD+SHE are still not training-agnostic.

ResNet-18 and ResNet-34 resutls on CIFAR-10 trained with Supervised Contrastive Learning are shown in Table 4, while Table 5 shows on CIFAR-10 when trained with cross entropy.

Table 6 shows $\text{HAC}_k$-OOD is not compatible with ReAct as an activation clipping method. We believe this is due to $\text{HAC}_k$-OOD already similarly limiting the expansion of the class contour when calculating the radial boundary. On all experiments 0.95 was used as the clipping quantile.

### A.5 ALGORITHMS

Algorithm 1 details hypercone construction for ID data contouring while 2 details OOD inference.

Table 3: *Far-OOD detection* CIFAR-100 Cross Entropy Loss.

| Backbone | Method | Textures FPR95 ↓ | Textures AUROC ↑ | iSUN FPR95 ↓ | iSUN AUROC ↑ | LSUN FPR95 ↓ | LSUN AUROC ↑ | Places365 FPR95 ↓ | Places365 AUROC ↑ | SVHN FPR95 ↓ | SVHN AUROC ↑ | Average FPR95 ↓ | Average AUROC ↑ |
|---|---|---|---|---|---|---|---|---|---|---|---|---|---|
| ResNet-18 | MSP | 86.95 | 73.22 | 83.92 | 71.6 | 82.61 | 78.09 | 83.96 | 74.32 | 88.61 | 68.67 | 85.21 | 73.18 |
| | MaxLogit | 86.61 | 73.78 | 81.15 | 76.13 | 80.65 | 81.25 | 83.37 | 75.10 | 87.53 | 69.80 | 83.86 | 75.21 |
| | Energy | 86.37 | 73.67 | 79.54 | 76.77 | 80.55 | 81.55 | 84.11 | 75.03 | 87.33 | 69.90 | 83.58 | 75.38 |
| | ViM | 58.74 | 86.01 | 86.57 | 75.06 | 88.4 | 74.37 | 86.98 | 71.58 | 80.46 | 79.55 | 80.23 | 77.31 |
| | GradNorm | 86.22 | 74.41 | 79.81 | 74.95 | 80.19 | 80.63 | 83.67 | 75.34 | 87.68 | 69.55 | 83.51 | 74.98 |
| | Mahalanobis | 65.35 | 80.50 | 93.29 | 62.05 | 95.09 | 61.97 | 93.49 | 61.29 | 80.86 | 77.29 | 85.62 | 68.62 |
| | KL Matching | 82.73 | 74.06 | 82.98 | 71.33 | 79.91 | 78.29 | 82.96 | 74.10 | 84.27 | 69.78 | 82.57 | 73.51 |
| | KNN+ | 71.35 | 79.10 | 91.57 | 65.42 | 91.50 | 65.76 | 91.45 | 64.94 | 89.9 | 71.95 | 87.15 | 69.43 |
| | SSD+ | 72.73 | 71.55 | 95.78 | 50.59 | 98.86 | 41.84 | 96.83 | 45.89 | 87.94 | 65.49 | 90.43 | 55.07 |
| | GEN | 86.54 | 73.84 | 79.69 | 76.62 | 80.53 | 81.48 | 83.99 | 75.11 | 87.48 | 69.87 | 83.65 | 75.38 |
| | NNGuide | 85.57 | 75.5 | 78.11 | 78.11 | 79.54 | 81.37 | 83.12 | 75.17 | 87.27 | 70.46 | 82.72 | 76.12 |
| | SHE | 85.34 | 69.4 | 82.01 | 76.43 | 78.73 | 81.15 | 85.2 | 71.59 | 81.95 | 70.19 | 82.65 | 73.75 |
| | ASH | 56.44 | 85.02 | 89.24 | 66.35 | 58.54 | 85.68 | 89.22 | 66.56 | 47.71 | 89.05 | 68.23 | 78.53 |
| | SCALE | 58.99 | 85.96 | 79.74 | 79.13 | 45.81 | 91.22 | 85.60 | 74.29 | 53.94 | 88.41 | 64.82 | 83.80 |
| | HAC$_k$-OOD | 67.38 | 81.98 | 87.05 | 70.19 | 81.76 | 78.38 | 84.88 | 72.98 | 82.15 | 78.64 | 80.64 | 76.43 |
| | Mahalanobis+SHE | 65.37 | 80.50 | 93.30 | 62.05 | 95.10 | 61.97 | 93.49 | 61.29 | 80.87 | 77.29 | 85.63 | 68.62 |
| | HAC$_k$-OOD+SHE | 67.39 | 82.01 | 87.04 | 70.24 | 81.73 | 78.43 | 84.89 | 73.02 | 82.26 | 78.5 | 80.66 | 76.44 |
| ResNet-34 | MSP | 81.01 | 75.71 | 82.76 | 73.22 | 82.10 | 77.01 | 81.21 | 75.88 | 77.07 | 79.95 | 80.83 | 76.35 |
| | MaxLogit | 79.13 | 77.65 | 79.65 | 76.73 | 82.82 | 77.76 | 80.79 | 76.23 | 75.31 | 82.40 | 79.54 | 78.15 |
| | Energy | 78.83 | 77.92 | 77.05 | 77.43 | 84.64 | 77.54 | 80.84 | 76.14 | 74.80 | 82.72 | 79.23 | 78.35 |
| | ViM | 70.76 | 81.99 | 84.64 | 77.10 | 73.73 | 81.10 | 84.24 | 74.47 | 71.77 | 83.99 | 77.03 | 79.73 |
| | GradNorm | 78.32 | 77.45 | 77.14 | 76.31 | 82.55 | 78.14 | 79.93 | 76.62 | 73.91 | 82.26 | 78.37 | 78.16 |
| | Mahalanobis | 73.42 | 79.66 | 85.76 | 74.51 | 70.45 | 82.12 | 84.93 | 73.81 | 77.14 | 81.93 | 78.34 | 78.41 |
| | KL Matching | 79.66 | 75.72 | 81.22 | 73.15 | 80.21 | 76.81 | 80.66 | 75.67 | 74.59 | 80.24 | 79.27 | 76.32 |
| | KNN+ | 78.88 | 76.31 | 86.44 | 73.21 | 72.59 | 78.55 | 84.08 | 73.59 | 78.49 | 79.86 | 80.19 | 76.30 |
| | SSD+ | 82.41 | 70.44 | 91.57 | 62.85 | 88.13 | 70.22 | 93.18 | 58.61 | 89.19 | 69.44 | 88.90 | 66.31 |
| | GEN | 78.65 | 77.9 | 77.12 | 77.3 | 84.09 | 77.67 | 80.58 | 76.22 | 74.54 | 82.69 | 79.00 | 78.36 |
| | NNGuide | 76.86 | 78.49 | 77.22 | 77.86 | 77.56 | 80.1 | 79.21 | 77.01 | 72.47 | 83.34 | 76.66 | 79.36 |
| | SHE | 79.77 | 75.78 | 79.23 | 75.79 | 89.62 | 72.83 | 83.18 | 73.53 | 78.02 | 80.24 | 81.96 | 75.63 |
| | ASH | 74.52 | 77.4 | 81.10 | 73.22 | 83.09 | 77.99 | 79.97 | 75.72 | 70.33 | 82.86 | 77.8 | 77.78 |
| | SCALE | 74.95 | 78.98 | 80.85 | 73.28 | 82.72 | 78.44 | 79.42 | 76.09 | 70.12 | 83.03 | 77.61 | 77.96 |
| | HAC$_k$-OOD | 73.67 | 80.07 | 84.36 | 73.63 | 55.55 | 86.13 | 78.00 | 78.00 | 74.21 | 83.69 | 73.16 | 80.30 |
| | Mahalanobis+SHE | 73.46 | 79.67 | 85.77 | 74.51 | 70.52 | 82.12 | 84.97 | 73.81 | 77.18 | 81.93 | 78.38 | 78.41 |
| | HAC$_k$-OOD+SHE | 73.6 | 80.13 | 84.34 | 73.68 | 55.49 | 86.15 | 77.97 | 78.03 | 74.25 | 83.72 | 73.13 | 80.34 |
| ResNet-50 | MSP | 83.60 | 75.6 | 82.14 | 77.16 | 79.58 | 80.4 | 81.39 | 76.77 | 85.17 | 76.20 | 82.32 | 77.23 |
| | MaxLogit | 82.15 | 76.99 | 78.76 | 80.83 | 77.87 | 82.15 | 80.3 | 77.41 | 85.54 | 77.60 | 80.92 | 79.00 |
| | Energy | 81.97 | 77.12 | 76.17 | 81.42 | 77.99 | 82.24 | 80.53 | 77.40 | 86.48 | 77.55 | 80.63 | 79.15 |
| | ViM | 36.49 | 92.24 | 57.27 | 87.73 | 72.69 | 83.38 | 81.32 | 76.41 | 37.66 | 91.52 | 57.09 | 86.26 |
| | GradNorm | 82.34 | 76.90 | 76.99 | 80.05 | 77.93 | 81.75 | 80.32 | 77.76 | 86.19 | 77.42 | 80.75 | 78.78 |
| | Mahalanobis | 45.69 | 89.88 | 76.13 | 80.41 | 85.29 | 76.45 | 89.89 | 67.10 | 48.05 | 89.60 | 69.01 | 80.69 |
| | KL Matching | 80.67 | 76.18 | 80.24 | 77.20 | 76.80 | 80.78 | 80.59 | 76.68 | 81.05 | 76.77 | 79.87 | 77.52 |
| | KNN+ | 63.94 | 82.46 | 76.54 | 78.78 | 76.37 | 79.82 | 80.48 | 75.63 | 56.91 | 84.32 | 70.85 | 80.20 |
| | SSD+ | 51.4 | 86.05 | 80.16 | 74.87 | 90.43 | 66.04 | 92.94 | 57.02 | 55.39 | 86.24 | 74.06 | 74.04 |
| | GEN | 82.06 | 77.13 | 76.38 | 81.30 | 78.03 | 82.23 | 80.51 | 77.45 | 86.46 | 77.58 | 80.69 | 79.14 |
| | NNGuide | 80.43 | 77.84 | 74.62 | 81.86 | 76.69 | 82.38 | 79.71 | 77.83 | 83.07 | 79.1 | 78.9 | 79.80 |
| | SHE | 78.51 | 76.49 | 76.46 | 81.07 | 69.78 | 83.51 | 80.87 | 75.68 | 83.1 | 75.96 | 77.74 | 78.54 |
| | ASH | 55.43 | 85.74 | 78.11 | 71.82 | 37.1 | 93.22 | 80.1 | 73.53 | 53.61 | 88.63 | 60.87 | 82.59 |
| | SCALE | 56.15 | 85.26 | 79.8 | 70.4 | 35.52 | 93.39 | 79.34 | 73.68 | 53.08 | 88.64 | 60.90 | 82.27 |
| | HAC$_k$-OOD | 64.17 | 83.9 | 74.70 | 80.18 | 52.34 | 88.26 | 73.15 | 79.56 | 58.39 | 85.14 | 64.55 | 83.41 |
| | Mahalanobis+SHE | 46.69 | 89.89 | 76.13 | 80.41 | 85.29 | 76.45 | 89.89 | 67.09 | 48.04 | 89.60 | 69.01 | 80.69 |
| | HAC$_k$-OOD+SHE | 64.33 | 83.89 | 74.76 | 80.49 | 51.69 | 88.62 | 72.91 | 79.93 | 58.63 | 85.29 | 64.46 | 83.64 |

Table 4: *Far-OOD detection* CIFAR-10 with Supervised Contrastive Learning.

| Backbone | Method | Textures FPR95 ↓ | Textures AUROC ↑ | iSUN FPR95 ↓ | iSUN AUROC ↑ | LSUN FPR95 ↓ | LSUN AUROC ↑ | Places365 FPR95 ↓ | Places365 AUROC ↑ | SVHN FPR95 ↓ | SVHN AUROC ↑ | Average FPR95 ↓ | Average AUROC ↑ |
|---|---|---|---|---|---|---|---|---|---|---|---|---|---|
| ResNet-18 | MSP | 43.53 | 94.21 | 48.18 | 93.33 | 31.24 | 95.82 | 55.53 | 90.65 | 37.83 | 94.74 | 43.26 | 93.75 |
| | MaxLogit | 22.68 | 96.12 | 24.95 | 96.12 | 7.82 | 98.47 | 35.36 | 93.17 | 18.41 | 96.88 | 21.84 | 96.22 |
| | Energy | 21.49 | 96.6 | 23.47 | 96.32 | 6.81 | 98.66 | 33.87 | 93.35 | 17.55 | 97.02 | 20.64 | 96.39 |
| | ViM | 15.62 | 97.53 | 53.76 | 93.01 | 3.83 | 99.00 | 28.64 | 94.47 | 1.00 | 99.78 | 20.57 | 96.76 |
| | GradNorm | 35.11 | 94.88 | 40.03 | 93.98 | 22.75 | 96.73 | 48.26 | 91.18 | 29.74 | 95.47 | 35.18 | 94.45 |
| | Mahalanobis | 14.06 | 97.75 | 57.90 | 92.45 | 3.32 | 99.16 | 27.06 | 94.77 | 0.63 | 99.85 | 20.59 | 96.80 |
| | KL Matching | 44.66 | 93.42 | 47.70 | 92.52 | 30.26 | 95.79 | 55.00 | 88.40 | 43.45 | 93.96 | 43.45 | 92.82 |
| | KNN+ | 14.66 | 97.68 | 28.07 | 95.68 | 3.38 | 99.27 | 28.87 | 94.51 | 3.47 | 99.39 | 15.69 | 97.31 |
| | SSD+ | 14.57 | 97.70 | 64.40 | 91.39 | 3.54 | 99.13 | 28.23 | 94.65 | 0.55 | 99.87 | 22.26 | 96.55 |
| | GEN | 23.42 | 96.32 | 26.48 | 95.88 | 9.04 | 98.37 | 36.55 | 92.92 | 18.99 | 96.80 | 22.90 | 96.06 |
| | NNGuide | 20.02 | 96.91 | 25.8 | 95.97 | 6.52 | 98.74 | 34.93 | 93.12 | 13.55 | 97.71 | 20.16 | 96.49 |
| | SHE | 14.5 | 97.57 | 30.02 | 95.2 | 3.55 | 99.32 | 32.37 | 93.65 | 4.83 | 99.17 | 17.05 | 96.98 |
| | ASH | 11.37 | 97.05 | 28.08 | 91.53 | 1.90 | 99.46 | 33.78 | 88.92 | 11.86 | 96.79 | 17.40 | 94.75 |
| | SCALE | 12.45 | 97.23 | 25.52 | 95.83 | 1.89 | 99.49 | 33.39 | 90.33 | 14.49 | 96.08 | 17.55 | 95.39 |
| | HAC$_k$-OOD | 16.45 | 97.13 | 26.26 | 95.86 | 3.27 | 99.36 | 26.91 | 94.77 | 6.02 | 98.95 | 15.78 | 97.21 |
| | Mahalanobis+SHE | 14.08 | 97.75 | 57.89 | 92.45 | 3.32 | 99.16 | 27.06 | 94.77 | 0.63 | 99.85 | 20.60 | 96.80 |
| | HAC$_k$-OOD+SHE | 15.37 | 97.29 | 27.37 | 95.61 | 3.5 | 99.29 | 27.64 | 94.53 | 5.76 | 98.95 | 15.93 | 97.13 |
| ResNet-34 | MSP | 28.24 | 95.8 | 23.71 | 96.45 | 11.31 | 97.94 | 39.04 | 92.99 | 13.03 | 97.73 | 23.07 | 96.18 |
| | MaxLogit | 18.62 | 96.76 | 9.73 | 98.09 | 2.40 | 99.21 | 25.25 | 94.94 | 5.09 | 98.83 | 12.22 | 97.57 |
| | Energy | 18.81 | 96.80 | 9.65 | 98.16 | 2.30 | 99.29 | 25.17 | 94.99 | 5.02 | 98.88 | 12.19 | 97.62 |
| | ViM | 10.32 | 98.11 | 13.82 | 97.34 | 1.95 | 99.15 | 23.08 | 95.35 | 0.81 | 99.84 | 10.00 | 97.96 |
| | GradNorm | 28.24 | 96.12 | 23.71 | 96.84 | 11.31 | 98.42 | 39.04 | 93.26 | 13.03 | 98.14 | 23.07 | 96.56 |
| | Mahalanobis | 8.65 | 98.50 | 14.18 | 97.38 | 1.99 | 99.39 | 21.54 | 95.58 | 0.64 | 99.88 | 9.40 | 98.15 |
| | KL Matching | 28.30 | 93.91 | 23.28 | 95.05 | 10.71 | 97.82 | 38.16 | 89.84 | 13.25 | 97.27 | 22.74 | 94.78 |
| | KNN+ | 13.67 | 97.84 | 13.99 | 97.58 | 2.40 | 99.4 | 24.37 | 95.23 | 2.70 | 99.47 | 11.43 | 97.90 |
| | SSD+ | 8.17 | 98.55 | 15.68 | 97.23 | 1.59 | 99.39 | 22.72 | 95.50 | 0.54 | 99.89 | 9.74 | 98.11 |
| | GEN | 19.08 | 96.81 | 10.97 | 97.98 | 3.05 | 99.18 | 26.57 | 94.78 | 5.70 | 98.79 | 13.07 | 97.51 |
| | NNGuide | 17.59 | 97.13 | 11.16 | 97.99 | 3.15 | 99.15 | 27.31 | 94.71 | 7.34 | 98.56 | 13.31 | 97.5 |
| | SHE | 13.67 | 97.63 | 17.88 | 96.94 | 2.57 | 99.46 | 27.62 | 94.31 | 2.88 | 99.41 | 12.92 | 97.55 |
| | ASH | 10.35 | 98.24 | 16.65 | 97.02 | 2.55 | 99.31 | 26.16 | 94.32 | 11.11 | 97.91 | 13.36 | 97.36 |
| | SCALE | 10.80 | 98.15 | 14.57 | 97.30 | 2.52 | 99.28 | 25.52 | 94.49 | 11.37 | 97.76 | 12.96 | 97.40 |
| | HAC$_k$-OOD | 14.31 | 97.45 | 12.44 | 97.59 | 2.03 | 99.50 | 20.84 | 95.41 | 3.32 | 99.33 | 10.59 | 97.86 |
| | Mahalanobis+SHE | 8.65 | 98.50 | 14.18 | 97.38 | 1.99 | 99.39 | 21.54 | 95.58 | 0.64 | 99.88 | 9.40 | 98.15 |
| | HAC$_k$-OOD+SHE | 13.85 | 97.68 | 13.32 | 97.59 | 2.02 | 99.53 | 21.96 | 95.3 | 3.5 | 99.33 | 10.93 | 97.89 |

Table 5: *Far-OOD detection* CIFAR-10 with Cross Entropy Loss.

| Backbone | Method | Textures | | iSUN | | LSUN | | Places365 | | SVHN | | Average | |
|---|---|---|---|---|---|---|---|---|---|---|---|---|---|
| | | FPR95 ↓ | AUROC ↑ | FPR95 ↓ | AUROC ↑ | FPR95 ↓ | AUROC ↑ | FPR95 ↓ | AUROC ↑ | FPR95 ↓ | AUROC ↑ | FPR95 ↓ | AUROC ↑ |
| ResNet-18 | MSP | 58.49 | 89.66 | 46.11 | 93.2 | 40.26 | 93.81 | 58.88 | 88.2 | 49.99 | 92.79 | 50.75 | 91.53 |
| | MaxLogit | 52.7 | 89.91 | 36.46 | 94.04 | 28.68 | 95.04 | 52.3 | 88.76 | 41.14 | 93.53 | 42.26 | 92.26 |
| | Energy | 51.15 | 89.98 | 34.45 | 94.16 | 26.67 | 95.19 | 50.75 | 88.84 | 39.01 | 93.62 | 40.41 | 92.36 |
| | ViM | **25.39** | **95.32** | **24.56** | **95.87** | **20.89** | **96.42** | **46.39** | **89.98** | **19.66** | **96.75** | **27.38** | **94.87** |
| | GradNorm | 54.86 | 90 | 41.9 | 93.68 | 35.96 | 94.38 | 55.86 | 88.53 | 45.1 | 93.2 | 46.74 | 91.96 |
| | Mahalanobis | 28.99 | 94.53 | 31.54 | 94.75 | 27.32 | 95.52 | 50.32 | 88.72 | 29.96 | 95.24 | 33.63 | 93.75 |
| | KL Matching | 58.97 | 89.25 | 47.06 | 92.82 | 41.09 | 93.37 | 59.19 | 86.98 | 50.76 | 92.44 | 51.41 | 90.97 |
| | KNN+ | 50.28 | 91.80 | 41.52 | 94.03 | 34.68 | 95.02 | 53.23 | 89.26 | 44.73 | 93.87 | 44.89 | 92.80 |
| | SSD+ | 32.36 | 93.67 | 49.64 | 91.80 | 38.31 | 94.14 | 65.57 | 85.39 | 33.62 | 94.55 | 43.9 | 91.91 |
| | GEN | 52.02 | 90.04 | 35.98 | 94.1 | 28.57 | 95.06 | 51.93 | 88.81 | 40.34 | 93.56 | 41.77 | 92.31 |
| | NNGuide | 51.95 | 90.53 | 35.36 | 94.31 | 29.03 | 94.9 | 51.27 | 89.17 | 41.66 | 93.4 | 41.85 | 92.46 |
| | SHE | 55.74 | 88.92 | 36.85 | 93.75 | 29.52 | 94.73 | 54.44 | 87.38 | 43.63 | 93.06 | 44.04 | 91.57 |
| | ASH | 57.06 | 88.18 | 38.87 | 93.65 | 35.69 | 93.54 | 57.16 | 87.35 | 48.76 | 92.21 | 47.51 | 90.99 |
| | SCALE | 55.71 | 88.70 | 38.38 | 93.81 | 33.66 | 94.04 | 55.93 | 87.77 | 47.21 | 92.60 | 46.18 | 91.38 |
| | HAC$_k$-OOD | 63.01 | 87.91 | 41.23 | 93.09 | 48.85 | 91.32 | 53.79 | 87.05 | 68.2 | 89.51 | 55.02 | 89.78 |
| | Mahalanobis+SHE | 28.99 | 94.53 | 31.54 | 94.75 | 27.32 | 95.52 | 50.32 | 88.72 | 29.96 | 95.24 | 33.63 | 93.75 |
| | HAC$_k$-OOD+SHE | 66.84 | 86.04 | 45.13 | 91.95 | 53.36 | 89.75 | 56.29 | 85.41 | 74.23 | 87.56 | 59.17 | 88.14 |
| ResNet-34 | MSP | 63.3 | 87.67 | 61.06 | 89.86 | 43.8 | 93.34 | 63.41 | 87.33 | 54.42 | 92.7 | 57.2 | 90.18 |
| | MaxLogit | 56.91 | 87.94 | 52.69 | 90.52 | 31.55 | 94.54 | 55.23 | 87.9 | 43.95 | 93.61 | 48.07 | 90.9 |
| | Energy | 56.26 | 88.01 | 51.78 | 90.6 | 30.39 | 94.67 | 54.33 | 87.98 | 42.52 | 93.72 | 47.06 | 91.00 |
| | ViM | 33.83 | 92.87 | **41.77** | 92.87 | **16.32** | **97.02** | 50.03 | 90.18 | 23.74 | 96.19 | 33.14 | 94.01 |
| | GradNorm | 63.3 | 87.84 | 61.06 | 90.08 | 43.8 | 93.77 | 63.41 | 87.53 | 54.42 | 92.98 | 57.2 | 90.44 |
| | Mahalanobis | 44.02 | 93.14 | 50.63 | 92.41 | 30.76 | 95.86 | 55.89 | 89.94 | 41.24 | 94.68 | 44.51 | 93.21 |
| | KL Matching | 63.53 | 86.73 | 61.37 | 88.41 | 44.38 | 92.84 | 63.44 | 85.92 | 54.88 | 92.21 | 57.52 | 89.22 |
| | KNN+ | 58.37 | 90.3 | 55.94 | 90.79 | 37.12 | 94.92 | 57.31 | 89.38 | 50.53 | 93.29 | 51.85 | 91.74 |
| | SSD+ | **29.59** | **94.76** | 41.95 | **93.36** | 17.04 | 96.99 | 53.04 | **90.26** | **17.28** | **96.92** | **31.78** | **94.46** |
| | GEN | 57.36 | 87.98 | 53.47 | 90.49 | 32.91 | 94.49 | 56.13 | 87.88 | 45.01 | 93.55 | 48.98 | 90.88 |
| | NNGuide | 57.32 | 88.61 | 52.68 | 91.04 | 32.89 | 94.64 | 55.22 | 88.65 | 46.01 | 93.32 | 48.82 | 91.25 |
| | SHE | 57.93 | 87.93 | 54.38 | 90.49 | 34.27 | 94.11 | 59.27 | 86.4 | 42 | 94.06 | 49.57 | 90.6 |
| | ASH | 58.74 | 87.82 | 56.04 | 90.24 | 38.15 | 93.85 | 60.69 | 86.89 | 46.42 | 93.68 | 52.01 | 90.50 |
| | SCALE | 58.33 | 87.88 | 54.97 | 90.35 | 35.72 | 94.13 | 58.93 | 87.22 | 45.53 | 93.69 | 50.70 | 90.65 |
| | HAC$_k$-OOD | 77.46 | 78.14 | 68.18 | 82.69 | 62.6 | 85.83 | 67.95 | 81.3 | 85.84 | 79.01 | 72.41 | 81.39 |
| | Mahalanobis+SHE | 44.02 | 93.14 | 50.63 | 92.41 | 30.76 | 95.86 | 55.89 | 89.94 | 41.24 | 94.68 | 44.51 | 93.21 |
| | HAC$_k$-OOD+SHE | 76.95 | 78.36 | 67.46 | 82.89 | 61.96 | 86.02 | 67.4 | 81.48 | 85.32 | 79.25 | 71.82 | 81.6 |

Table 6: CIFAR-100 Supervised Contrastive Learning ReAct Ablation.

| Backbone | Method | Textures | | iSUN | | LSUN | | Places365 | | SVHN | | Average | |
|---|---|---|---|---|---|---|---|---|---|---|---|---|---|
| | | FPR95 ↓ | AUROC ↑ | FPR95 ↓ | AUROC ↑ | FPR95 ↓ | AUROC ↑ | FPR95 ↓ | AUROC ↑ | FPR95 ↓ | AUROC ↑ | FPR95 ↓ | AUROC ↑ |
| ResNet-18 | HAC$_k$-OOD | **52.7** | **88.03** | 75.91 | 83.52 | **31.89** | **93.52** | **76.30** | **79.71** | 52.95 | 89.78 | **57.95** | **86.91** |
| | HAC$_k$-OOD +React | 88.23 | 77.1 | **67.44** | **85.28** | 88.41 | 68.87 | 78.36 | 78.54 | 69.00 | 85.92 | 78.29 | 79.14 |
| ResNet-34 | HAC$_k$-OOD | **46.17** | **90.08** | **61.14** | **87.47** | **33.17** | **93.61** | **71.31** | 81.29 | 29.01 | 94.47 | **48.16** | **89.38** |
| | HAC$_k$-OOD +React | 89.13 | 80.01 | 72.25 | 85.91 | 80.3 | 85.28 | 75.98 | 80.92 | 38.81 | 93.65 | 71.29 | 85.15 |
| ResNet-50 | HAC$_k$-OOD | **34.29** | **92.95** | **63.13** | 86.72 | **15.86** | **97.12** | **71.93** | 82.12 | **14.66** | **97.09** | **39.97** | **91.2** |
| | HAC$_k$-OOD +React | 94.77 | 69.34 | 86.23 | 77.8 | 84.92 | 80.66 | 72.83 | **82.29** | 51.28 | 91.84 | 78.01 | 80.39 |

---

**Algorithm 1** Hypercone Construction for ID Data Contouring.

---

**Input:** $X_{\text{train}}, X_{\text{test}}, f_{\text{encoder}}, Y, \text{normalized}$
**Output:** $H, C, \lambda$

1: **Function** ExtractEmbeddings$(X, f_{\text{encoder}})$: return $f_{\text{encoder}}(X)$
2: **Function** GetObsAtClass$(Z, l)$: return features corresponding to class $l$
3: **Function** NN$(C_l, Z)$: return nearest neighbor to $C_l$ among points in $Z$
4: **Function** KNNAngle$(\vec{\alpha}, Z)$: return cosine distance to $k$-th nearest neighbor of $\vec{\alpha}$ in cosine distance among points in $Z$
5: $Z_{train} = f_{\text{encoder}}(X_{\text{train}})$
6: $Z_{test} = f_{\text{encoder}}(X_{\text{test}})$
7: $C = \left\{ \frac{1}{|Z_{train_l}|} \sum_{i=1}^{|Z_{train_l}|} z_{train_{l,i}} \ \ \forall l \in Y \right\}$ (compute class centroids)
8: **for** $l \in Y$ **do**
9: $\quad Z_{train_l} = $ GetObsAtClass$(Z_{train}, l)$
10: $\quad Z_{test_l} = $ GetObsAtClass$(Z_{test}, l)$
11: $\quad$ **if** normalized **then**
12: $\qquad C_l = $ NN$(C_l, Z_{train_l})$
13: $\quad Z_{train_l} = \{z - C_l \ \forall \ z \in Z_{train_l}\}$ (center train embeddings at centroid)
14: $\quad Z_{test_l} = \{z - C_l \ \forall \ z \in Z_{test_l}\}$ (center test embeddings at centroid)
15: $\quad A_l = \{\overrightarrow{C_l z} \ \forall \ z \in Z_{train_l}\}$ (compute hypercone axes)
16: $\quad T_l = \{\text{KNNAngle}(\vec{a_j}, Z_{train_l}) \ \forall \ j \in \{1, \dots, |A_l|\}\}$ (compute hypercone opening angles)
17: $\quad H_l = \{h(\vec{a_j}, \theta_j) \ \forall \ j \in \{1, \dots, |A_l|\}\}$ (construct hypercones)
18: $\quad$ **for** $h_{l,i} \in H_l$ **do**
19: $\qquad G_{l,i} = \{z \ \forall \ z \in Z_{train_l} \cup Z_{test_l} \mid \tau < \theta_{l,i}\}$ (identify observations in hypercone)
20: $\qquad D_{l,i} = \{\overrightarrow{|C_l g|} \ \forall \ g \in G_{l,i}\}$ (compute distances from centroid)
21: $\qquad b_{l,i} = \mu(D_{l,i}) + 2\sigma(D_{l,i})$ (compute distance aware radial boundary)
22: $\qquad D_{l,i}^{norm} = \left\{ \frac{d}{b_{l,i}} \ \forall \ d \in D_{l,i} \right\}$ (scale distances by radial boundary)
23: $\lambda = $ 95-th percentile$(D^{norm})$
24: **return** $H, C, \lambda$

---

---

**Algorithm 2** OOD Inference

---

**Input:** $X_{\text{new}}, f_{\text{encoder}}, Y, H, C, \lambda$
**Output:** $ID$

1: **Function** InHyperconeAngular$(z, \vec{\alpha_{l,i}}, \theta_{l,i})$: return $\arccos\left( \frac{\vec{\alpha_{l,i}} \cdot z}{\|\vec{\alpha_{l,i}}\| \|z\|} \right) < \theta_{l,i}$ (indicator for whether $z$ is within the angular boundary of $h_{l,i}$ parameterized by $\vec{\alpha_{l,i}}$ and $\theta_{l,i}$)
2: **Function** InHyperconeRadial$(z, C_l, \lambda)$ : return $\|\overrightarrow{C_l z}\| < \lambda$ (indicator for whether $z$ is within the radial boundary)
3: $Z_{\text{new}} = f_{\text{encoder}}(X_{\text{new}})$
4: $ID = \{0\}^{|Z_{\text{new}}|}$ (initialize ID indicator vector)
5: **for** $l \in Y$ **do**
6: $\quad Z_{\text{new}_l} = \{z - C_l \ \forall z \in Z_{\text{new}}\}$ (center embeddings at centroid)
7: $\quad$ **for** $z_j \in Z_{\text{new}_l}$ **do**
8: $\qquad$ **if** $ID_j = 0$ and $\exists \ h_{l,i} \in H_l$ s.t. InHyperconeAngular$(z_j, \vec{\alpha_{l,i}}, \theta_{l,i})$ and InHyperconeRadial$(z_j, C_l, \lambda)$ **then**
9: $\qquad\quad ID_j = 1$ (if $z_j$ is in at least one hypercone's angular and radial boundaries for one label $Y$, it is ID)
10: **return** $ID$

---

