# HYPERCONE ASSISTED CONTOUR GENERATION FOR OUT-OF-DISTRIBUTION DETECTION

## 1 TIME COMPLEXITY

### 1.1 THEORETICAL ANALYSIS

Here, we will describe the time complexity of $HAC_k$-OOD, both in the training/hypercone construction phase and in the inference phase. Overall, both phases are efficient in terms of time complexity. The hypercone construction phase scales with $\mathcal{O}(n^2)$, while the inference phase scales with $\mathcal{O}(n)$, where $n$ represents the number of training observations. Neither phase is prohibitively expensive.

We will elaborate more upon each component of both phases. Let us use the following definitions:

$$d = \text{Dimension of the embedding space}$$
$$l = \text{Number of labels}$$
$$n = \text{Number of observations in the training data}$$
$$m = \text{Number of observations in the testing data}$$
$$\max_k = \text{Largest k used in hypercone angle construction}$$
$$p = \text{Number of observations in the potentially OOD inference data}$$

In practice, it is often that case that $n >> d, l, n, m, \max_k$. In theory, $p$ could contain infinitely many points, but we assume that the inference is performed on sime finite set of $p$ points, where $p << n$.

During hypercone construction, the following tasks are performed with the following associated time complexities. For larger modules, we mention sub-components which inform the overall time complexity. Note that the time complexity for each module is determined based on the dominant time complexities for the sub-components. Items marked with a (*) are optional.

1. Normalize embeddings (*): $\mathcal{O}(md + nd)$
   (a) Normalize train embeddings: $\mathcal{O}(nd)$
   (b) Normalize test embeddings: $\mathcal{O}(md)$
2. Filter out incorrectly classified predictions (*): $\mathcal{O}(n)$
3. Create label centroids: $\mathcal{O}(n \log(n) + nd)$
   (a) Get unique labels: $\mathcal{O}(n \log(n))$
   (b) Calculate centroid: $\mathcal{O}(nd)$
4. Replace label centroids with nearest neighbor in the data (*): $\mathcal{O}(n \log(nd))$ (in theory, this could be performed in $\mathcal{O}(n)$ with a more efficient algorithm, which we will explore in the future)
5. Choose $k$ for each label: $\mathcal{O}(nd + n \log(d \frac{n}{l}))$
   (a) Calculate embedding label statistics: $\mathcal{O}(nd)$
   (b) Generate uniformly distributed clusters and compute centroids: $\mathcal{O}(ld\max_k)$
   (c) Compute $\max_k$ nearest neighbors for training distribution: $\mathcal{O}(n \log(d \frac{n}{l}))$
   (d) Compute $\max_k$ nearest neighbors for uniform distribution: $\mathcal{O}(l\max_k \log(d\max_k))$
   (e) Compute ratio of average of distances to $k$-th nearest neighbor for 10 $k$'s for training distribution and average of distances to $k$-th nearest neighbor for 10 $k$'s for uniform distribution: $\mathcal{O}(n + \max_k)$

6. Get hypercone angles for each label: $\mathcal{O}(n)$

7. Build hypercones: $\mathcal{O}(d\frac{n}{l}(n+m))$

   (a) Construct hypercone axes: $\mathcal{O}(nd)$

   (b) Center and normalize test and train embeddings: $\mathcal{O}(nd + md)$

   (c) Identify which test and train embeddings are contained within each hypercone's angular boundary by computing the dot product between axes and embeddings: $\mathcal{O}(d\frac{n}{l}(n+m))$

   (d) Compute preliminary cone height for each hypercone based on statistics of test and train embeddings contained within each hypercone: worst cast $\mathcal{O}(n(\frac{n+m}{l}))$ but in likely case $\mathcal{O}(nc_1)$ where $c_1 << \frac{n+m}{l}$ represents average number of test and train points contained within each hypercone

8. Get scores for ID data: $\mathcal{O}(ndm)$

   (a) Identify which test embeddings are contained within each hypercone's angular boundary by computing the dot product between axes and embeddings: $\mathcal{O}(dnm)$

   (b) Normalize test embeddings: $\mathcal{O}(dm)$

   (c) For each hypercone, normalize distance between each contained test point and the centroid by the preliminary cone height: worst case $\mathcal{O}(mn)$ but in likely case $\mathcal{O}(nc_2)$ where $c_2 << m$ is the average number of test points contained within each hypercone

9. Compute boundary of ID data by sorting scores for ID data and indexing at desired FPR: $\mathcal{O}(m\log(m))$

The largest limiting factor in the time complexity of hypercone construction is therefore in step 7, when we compute the dot product between hypercone axes and train and test points to identify which hypercones the points fall in to inform the preliminary radial boundaries. Assuming $n >> d, l, m$, the $\mathcal{O}(d\frac{n}{l}(n+m))$ time complexity is roughly $\mathcal{O}(n^2)$.

During inference, the following tasks are performed.

1. Get OOD scores (same as getting scores for ID data): $\mathcal{O}(ndp)$

2. Compare OOD scores to boundary of ID data: $\mathcal{O}(p)$

Again, the largest limiting factor stems from computing the dot product between hypercone axes and the inference data embeddings to determine which hypercones the inference points fall in. Assuming $n >> d, m$, the time complexity of the inference step is about $\mathcal{O}(n)$.

## 1.2 EXPERIMENTAL ANALYSIS

In Table 1, we report the inference times for each of the benchmark models and in Table 2 we show the time to construct hypercones. Note that the inference times for $HAC_k$-OOD differ slightly from those reported in the original manuscript. This variation is due to randomness and our deliberate effort to avoid running any processes concurrently this time.

## 2 FIGURES ON COMPARISON TO KNN+

Figures 1 and 2 show the inter and intra class variations which could have an effect on the performance of KNN+.

## 3 EFFECTS OF PARAMETERS ON PERFORMANCE

Figures 3 and 4 show how $HAC_k$-OOD's performance varies with angular and radial boundaries independently.

|  | Res18 Cifar100 | Res34 Cifar100 | Res50 Cifar100 |
|---|---|---|---|
| MSP | 0.0009 | 0.0009 | 0.0009 |
| Mahalanobis | 0.1256 | 0.1249 | 0.8590 |
| MaxLogit | 0.0009 | 0.0009 | 0.0009 |
| Energy | 0.0012 | 0.0012 | 0.0012 |
| Energy+React | 0.0025 | 0.0021 | 0.0056 |
| Residual | 0.0023 | 0.0021 | 0.0096 |
| GradNorm | 0.1163 | 0.1140 | 0.1152 |
| SSD+ | 0.0672 | 0.0650 | 0.5277 |
| ViM | 0.0026 | 0.0025 | 0.0097 |
| KL-Matching | 1.4816 | 1.4860 | 1.4788 |
| KNN+ | 2.0963 | 2.1029 | 8.3601 |
| GEN | 0.0015 | 0.0013 | 0.0015 |
| NNGuide | 0.6949 | 0.7213 | 0.9360 |
| SHE | 0.0102 | 0.0104 | 0.0130 |
| ASH | 0.0091 | 0.0097 | 0.0134 |
| SCALE | 0.0094 | 0.0058 | 0.0122 |
| $HAC_k$-OOD | 0.9976 | 0.9505 | 2.2488 |
| $HAC_k$-OOD +React | 1.0598 | 0.9918 | 2.2336 |
| $HAC_k$-OOD+SHE | 0.9981 | 0.9451 | 2.2232 |
| $HAC_k$-OOD +ASH | 0.9788 | 0.8771 | 2.1402 |

Table 1: Comparison of inference times per sample (ms) on ResNet architectures with Cifar100.

|  | Res18 Cifar100 | Res34 Cifar100 | Res50 Cifar100 |
|---|---|---|---|
| Total (s) | 17.0924 | 15.9830 | 30.6542 |
| Per Training Sample (ms) | 0.3418 | 0.3197 | 0.6131 |

Table 2: $HAC_k$-OOD Hypercone Construction times for different ResNet architectures with Cifar100.

Toy Example 1

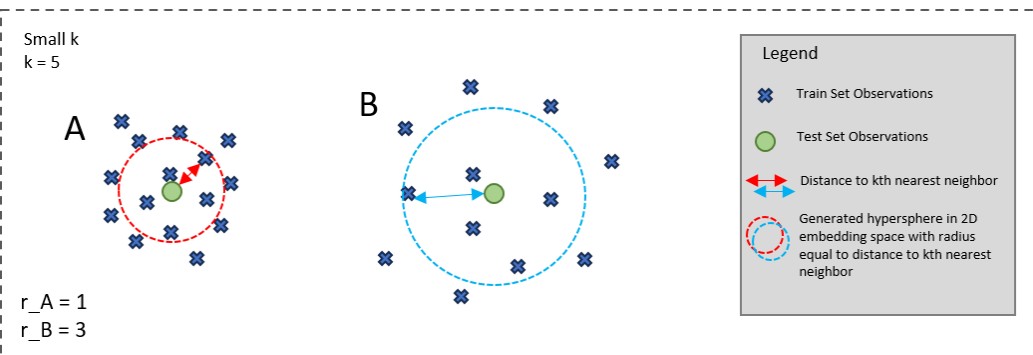

Figure 1: **Small k - Inter Cluster Density Variations:** KNN+ distances to 5th nearest neighbor is represented by hyperspheres in a 2D feature space. The radius of test set observation for class A is 1 and class B is 3.

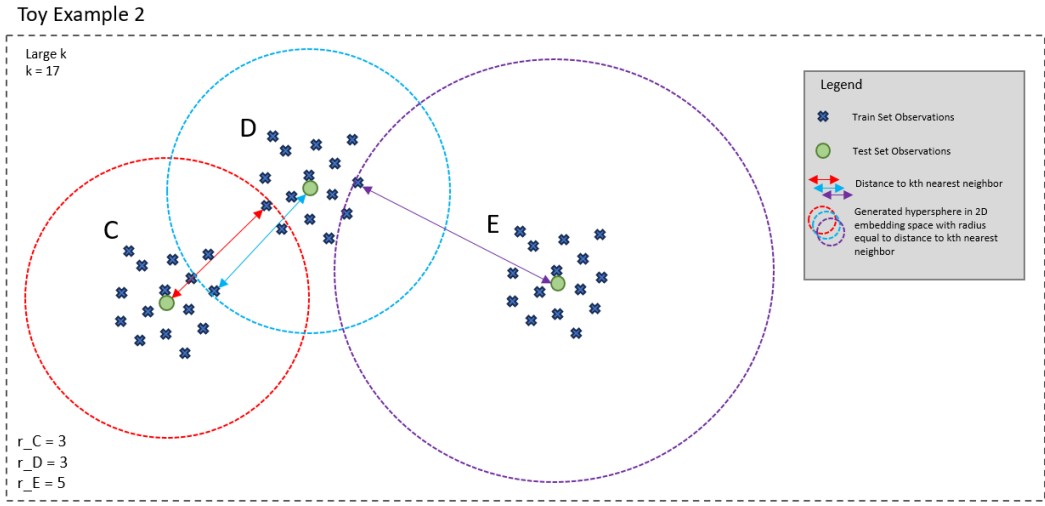

Figure 2: **Large k: Intra Cluster Distance Variations:** KNN+ distances to 17th nearest neighbor is represented by hyperspheres in a @D feature space. The radii of test set observation hyperspheres for classes C, D and E are 3, 3, and 5 respectively.

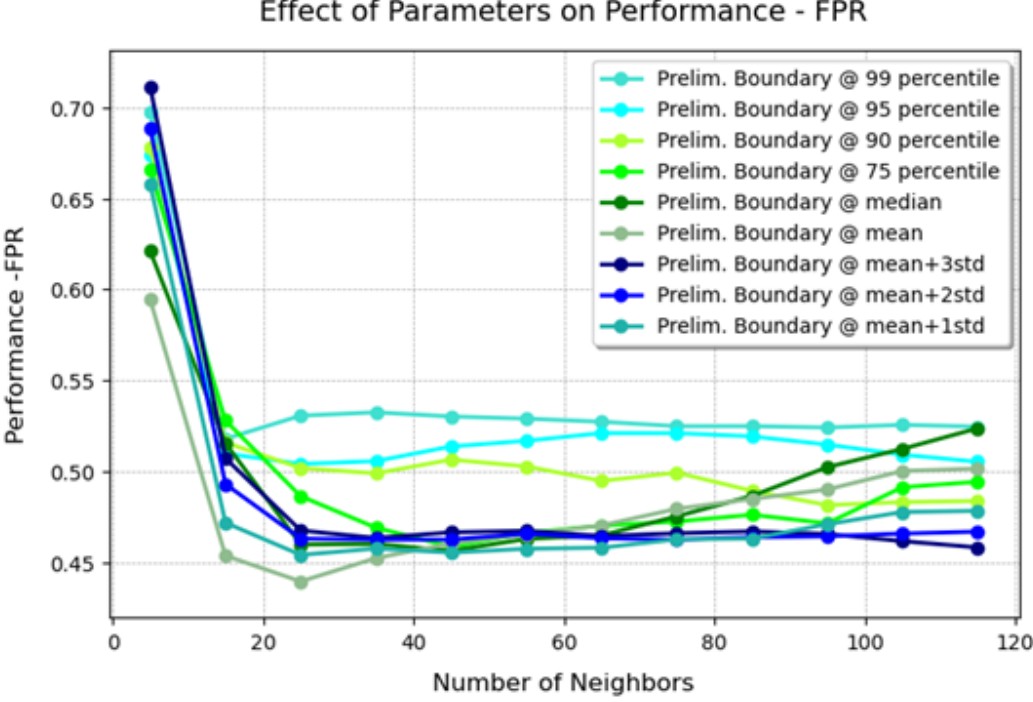

Figure 3: FPR95 performance of HAC$_k$-OOD when varying angular (number of neighbor) and radial (preliminary boundary) parameters.

216
217
218
219
220
221
222
223
224
225
226
227
228
229
230
231
232
233
234
235
236
237
238
239
240
241
242
243
244
245
246
247
248
249
250
251
252

Figure 4: AUROC performance of $\text{HAC}_k$-OOD when varying angular (number of neighbor) and radial (preliminary boundary) parameters.