# OpenReview forum: "Hypercone Assisted Contour Generation for Out-of-Distribution Detection"
_ICLR.cc/2025/Conference — Submitted to ICLR 2025_

### Official Review · Reviewer_TiVv · 2024-11-02

**Soundness:** 2
**Presentation:** 1
**Contribution:** 2
**Rating:** 5
**Confidence:** 3

**Summary:**

The paper presents a post-training strategy for Out-Of-Distribution (OOD) detection for image classification. The proposed method assumes no access to the OOD samples and employs a set of hypercones with varying cutoff distances in feature space to define the class boundaries of in-distribution data. The work evaluates this method and a combination with a previous OOD technique on the Far-OOD and near-OOD detection benchmarks, with comparisons to a set of baselines.

**Strengths:**

1. The paper addresses an important problem and proposes a new distance-based model based on hypercones.
2. The evaluation of the method seems comprehensive and it achieves strong results in some cases.

**Weaknesses:**

1. The paper is not clearly motivated. The discussion on training-based and distance-based post-training methods are insufficient. While the introduction section listed many previous methods, it is unclear what OOD modeling challenges this method aims to address, in particular for the distance-based approach.
2. The assumption of this method is very restrictive. As stated in Line 064, it requires "that ID and OOD data are separable in the space", which is unrealistic for real-world data.
3. The novelty of this method is limited. The proposed hypercone representation is similar to a mixture of Gaussian kernels for the ID data distribution.
4. The presentation of this work lacks clarity and the technical details are difficult to follow. Several parts of Sec 4.3 are confusing: 1) Line 260: Why do the hypercone representations rely on the test data feature Z_{test}, which should not be used during model construction? 2) Line 299: How is the score function defined and what threshold is used during the inference (in Sec 4.4)?
5. The experimental evaluation is lacking in three aspects: 1) The experimental setup is limited, which only considers three ResNet-based backbones. More modern architectures, such as ViT, should be included to validate its generalization. 2) The ablative study is lacking. What contributions are from the hypercones? What if it is replaced by Gaussians? 3) The performance of the original version HAC_k-OOD is mixed in both Table 1 and 2, and in most of cases, it is worse than the SOTA methods. It is unclear whether the proposed representation is truly effective.

**Questions:**

Please address the questions in the weaknesses part as above.

---

> ### Author Response · Authors · 2024-11-25
>
> Thank you for all your questions. Please find the responses below:
> 1. HAC provides a more flexible boundary for the ID space compared to other feature space methods. It can map complex spaces, by making no assumption on the distribution of the space. In contrast, SSD+ assumes a normal distribution and restricts class contours to hyperspheres. Additionally, SSD+ and KNN+ use a single global distance cutoff, while HAC allows for local cutoffs based on class statistics. A discussion on HAC's flexibility exists in the Methods section and we can include a note in the Introduction.
> 2. We use the word separable to describe an embedding space of a model that has been sufficiently trained to demonstrate superior performance in classification tasks. While regions of overlap between ID and OOD might exist, this would be an overall limitation of various distance based OOD detection methods. We would like to draw your attention to Table 2 in the paper, where HAC shows superior performance on Near-OOD, showing that it performs better than other methods in cases where ID and OOD classes overlap. We have clarified this in the paper and uploaded the updated version to OpenReview.
> 3. We appreciate the opportunity to clarify the novelty and distinctiveness of our hypercone approach in comparison to a mixture of Gaussian kernels.
>
>     (1)	HAC is grounded in the geometric properties of an embedding space, focusing on constructing explicit boundaries (contours) for OOD detection by using spatial and angular information in the data (cosine and Euclidean distances). The mixture of Gaussian kernels models the data distribution probabilistically by assuming the data can be represented as a combination of Gaussian distributions, focusing on the statistical properties of the data rather than the geometric structure.
>
>     (2)	At inference time, HAC computes whether a given point is within the computed boundary. However, a GMM approach computes the likelihood of a point being in distribution.
>
>     (3)	Gaussian kernels have also been used in open set recognition to shape embeddings into class-specific Gaussian distributions during training. HAC on the other hand is a post-training procedure. Thus, HAC is sufficiently different from a mixture of Gaussian kernels to be considered as novel – no other work has built hypercones to estimate the boundary of in distribution data.
>
> 4.
>
>    (1) We want to clarify that no OOD nor ID test data was used to construct the hypercones. Using classifier ID test set observations to determine the ID geometric boundary is standard practice in OOD detection literature. E.g., SSD forms a multidimensional hypersphere in Mahalanobis space as the ID contour and inflates it to include 95% of test set observations. Similarly, HAC uses training set observations to define hypercone directions, while test set observations determine which hypercones need inflating to classify 95% as ID.
>
>    (2) Please refer to lines 295–302. We will also elaborate upon the method with an example. Consider a single hypercone h_{0,0} projected from the centroid of cluster 0 with train set feature at position 0 defining its axis. The preliminary hypercone radial boundary is the mean plus two standard deviations of the feature lengths of observations within its angular boundary. If the preliminary length of h_{0,0} is 5 and that of h_{0,100} is 15, there is greater feature space variation in the h_{0,100} direction.  The threshold is the 95th percentile of test observation distances, normalized by the preliminary lengths of their respective hypercones, allowing for comparability across hypercones, For more details, see section 4.3.
>
> 5.
>
>    (1) Our experimental set-up is limited to architectures that are sensible for CIFAR-100. ViTs typically take as input bigger images than available in CIFAR and, although we could modify them to use smaller images, it would be hard to gather any meaningful insights from such experiments due to their size overfitting to them.
>
>    (2) Gaussian mixtures may work well when local regions are normally distributed but may fail in more irregular embedding spaces. HAC does not relying on unrealistic distributional assumptions, a global distance threshold or transformations of the embedding space. We didn't compare hypercones to a GMM, but we did test them against the Mahalanobis method which fits a univariate Gaussian for each class. HAC consistently outperformed Mahalanobis. We acknowledge the potential confusion in naming and will clarify in the final version.
>
>    (3) Tables 1 and 2 show that SOTA performance is achieved across various backbones and datasets when HAC and SHE are used together. SHE is a complementary method to HAC, and their combination offers a novel variant of HAC, differing only in centroid computation. By using only correctly classified data points to construct hypercones and define the ID contour, we mitigate issues from misclassified observations that may appear as outliers.

---

### Official Review · Reviewer_Fqvd · 2024-11-03

**Soundness:** 3
**Presentation:** 4
**Contribution:** 3
**Rating:** 6
**Confidence:** 4

**Summary:**

The paper introduces a post-training method for out-of-distribution (OOD) detection. The method approximates the contour of each class with a set of hypercones and defines per-hypercone decision boundaries.
More specifically, the hypercones are drawn as follows:
1. compute per-class centroids, which will be the apex of the class hypercones
2. take a class sample and set the hypercone axis to be the vector that points its (penultimate) representation
3. set the opening angle to be the angle between the hypercone axis and the k-th nearest neighbor from the sample representation
4. set the decision boundary using the distribution of representations within the hypercone's angular boundary

The method is an extension of another technique, SSD+, which assumes the decision boundary could be modeled with a unique hypersphere (or multidimensional ellipsoid) per class. It's training does not require OOD data.

Experimentally, the authors follow common practice and evaluate the OOD performance with the CIFAR-100 dataset as an in-distribution dataset and many different datasets as out-of-distribution datasets. Two types of pre-trained classification models are tested: models trained with a softmax cross-entropy loss and with a supervised contrastive loss.
The method achieves SOTA results in the supervised contrastive learning setup and is competitive in the cross-entropy setting.

**Strengths:**

1. The paper is well-written and easy to follow. The authors provide a good summary of the different approaches to OOD, a background section on hypercones, and a clear and precise method description.
2. Relevance and novelty of the method: the algorithm doesn't require assumptions about the data distribution and can model complex embedding spaces since it draws multiple hypercones per class and since it defines per-hypercone decision boundaries.
3. The authors discuss some limitations of the method (e.g., it works less well with smaller models ).

**Weaknesses:**

1. Limited evaluation: the method is only evaluated on models pre-trained on the *quite simple* CIFAR datasets and not on more complex datasets such as the ImageNet-200 or ImageNet-1k OOD benchmarks.

**Questions:**

- (see Weakness 1): how does the method perform on the ImageNet (200 or 1k) OOD benchmarks? Its evaluation on benchmarks beyond *quite simple* datasets (i.e., CIFAR-10/CIFAR-100) would strengthen the claims.
- What is the intuition behind the use of hypercones? Isn't the embedding space of a class more "dense" close to its centroid?

---

> ### Author Response · Authors · 2024-11-25
> **Hypercone Intuition**
>
> Please find our answers below:
> 1. Please refer to the rebuttal revision with regard to the Imagenet question.
> 2. Our intuition for using hypercones to map class contours stems from the need to extend the work of SSD+, which uses multidimensional ellipsoids to separate ID and OOD data in the embedding space. The hypercone approach aims to effectively map irregularly shaped clusters.
> We believe that the vast majority of data illustrate intrinsic hierarchical relationships. Given models large enough to capture these hierarchical structures we postulate that classes should typically deviate from Gaussian-like shapes. Specifically, clusters at the deepest level of the hierarchy are more likely to exhibit Gaussian-like shapes. However, as we move higher up the hierarchy, clusters tend to fuse together, becoming more irregular. HAC performs well even in classification frameworks which are not explicitly trained using hierarchical labels and we provide a justification on why that would be the case below:
> We draw your attention to two works [2, 3] that highlight the prevalence of hierarchical relationships in the vast majority of data. We hypothesize that HAC performs better as network sizes increase due to larger network ability to capture the irregular contours of clusters shaped by inherent hierarchies. Consider CIFAR-100, which includes a "fruit and vegetables" class. At the first level of the hierarchy, this class encompasses all relevant images. At the next level, it may split into two sub-clusters: "fruits" and "vegetables." The "fruits" sub-cluster could further divide into groups such as "citrus fruits" and "berry fruits," and at deeper levels, it might form clusters based on attributes like color. The larger the network the better the chance that it will be able to capture intricate relationships in data.
>
> [2] Erzsébet Ravasz and Albert-László Barabási. Hierarchical organization in complex networks. Physical Review E, 67(2), February 2003
>
> [3] Maximillian Nickel and Douwe Kiela. Poincaré embeddings for learning hierarchical representations. In Advances in Neural Information Processing Systems, 2017.

---

> ### Comment · Reviewer_Fqvd · 2024-12-02
>
> Dear authors,
>
> Thank you for the clarification on the intuition behind the use of hypercones to delimit the embedding space.
> After reviewing your rebuttal and considering the feedback and rebuttal to all four reviewers, I have decided to maintain the rating "marginally above the acceptance threshold".
> - **why not lower?** The approach is novel (it is similar to a mixture of Gaussian kernels but the use of hypercones is novel and is now motivated by some intuition). Additionally, the paper is easy to follow.
> - **why not higher?** Although I understand that HAC$_k$-OOD is not evaluated on ImageNet because of copyright concerns, the current training data is very limited in complexity. Specifically, CIFAR images are only 32x32 pixels, which restricts the ability to fully assess the approach's significance and applicability to real-world scenarios.

---

### Official Review · Reviewer_KcXV · 2024-11-04

**Soundness:** 3
**Presentation:** 3
**Contribution:** 3
**Rating:** 6
**Confidence:** 4

**Summary:**

The paper introduces a post-training out-of-distribution (OOD) detection method, HAC_k-OOD, which models the training data distribution through a set of hypercones and assesses OOD status based on whether a test sample falls within any hypercone. Specifically, for each class, the method first computes the class centroid and defines an angular boundary using the k-th nearest neighbors for each training point. Additionally, a radial boundary is set based on the mean and variance of the sample norms within the angular boundaries. During inference, a sample is classified as OOD if it lies outside either the angular or radial boundaries. Experiments were conducted using ResNet-18, ResNet-34, and ResNet-50, with both supervised contrastive learning and cross-entropy loss. The method was evaluated on CIFAR-10/100 as the in-distribution dataset and tested on various OOD datasets, covering both near and far OOD scenarios.

**Strengths:**

1. The method takes an interesting approach to distance-based OOD detection by relaxing the distributional assumptions and, unlike naive KNN, still leveraging nearby training data statistics to construct class contours. To the best of my knowledge, the use of hypercones for this purpose is novel and appears well-motivated.

2. The authors present their method clearly, making the paper easy to follow and understand.

3. Although the method involves hyperparameter k, the authors provide a practical approach to estimating it without requiring additional OOD data.

4. The experiments investigate various model sizes and training losses, demonstrating their impact on distance-based methods. Overall, the method benefits more from larger models trained with contrastive loss, as these models produce more distinguishable features.

**Weaknesses:**

1. The computational complexity is a concern for this method. Since the computation appears to increase with the size of the training dataset, it’s unclear if this approach would be feasible for large-scale, real-world applications. Although the authors state that the method is computationally efficient and support this with inference time per sample, I encourage them to provide a more detailed discussion on this aspect. For instance, what is the time required to construct the hypercones? A comparison of inference times with other methods would also be valuable.

2. How would the method perform on a large-scale, real-world dataset like ImageNet? Many recent OOD detection methods use ImageNet-1k as the in-distribution (ID) dataset. I encourage the authors to consider experiments on this dataset to evaluate the general applicability of the method in more realistic scenarios.

3. Recent work has explored OOD detection using CLIP as a backbone model (eg. [a1]), as CLIP may offer a more robust feature space. It would be interesting to see how this method performs when applied to a CLIP-based model.

4. Could the authors elaborate on why this method outperforms a naive KNN approach? One advantage seems to be that the method leverages the nearest neighbors within the training set (as opposed to KNN’s i.i.d. approach) to construct hypercones, which may capture more robust information about class boundaries. Additionally, an ablation study using either angular or radial boundaries separately for OOD detection could provide valuable insights into the method’s effectiveness and support future research.

5. While the paper is generally well-written, a few sections could be clearer. For example, in lines 80-81, P_in is referenced without being introduced in a previous formula. Additionally, in Section 5.2, only ResNet-34 is mentioned as the backbone model, though ResNet-18 and ResNet-50 are also used.

[a1] Ming, Yifei, et al. "Delving into out-of-distribution detection with vision-language representations." Advances in neural information processing systems 35 (2022).

**Questions:**

Please see the weaknesses part.

---

> ### Author Response · Authors · 2024-11-25
> **Complexity, CLIP, KNN+ and Parameter Ablations**
>
> Thank you for your insightful questions!
> 1. We have added an analysis of time complexity in the supplementary material uploaded to the rebuttal revision response, please refer to Section 1. We have also added run-times for both training and inference to tables 1 and 2. The inference times for HAC are consistent with the run times for other methods. Notably, the inference times per sample for both KNN+ and KL-Matching are significantly longer than for HAC. Furthermore, hypercone construction per training data point is fast, taking on average less than half a millisecond per sample across three architectures.
> 2. Please refer to the rebuttal revision response.
> 3. We agree that OOD detection on multimodal representations would be an interesting task to tackle. At the moment, it is out-of-scope for this work and we leave it open for future research.
> 4. We would like to clarify whether by "naive KNN" you are referring to KNN+ [1], an approach we regard highly for its simplicity and effectiveness. Analyzing KNN+ from a geometric perspective might help with comparison:
>
> Global vs. Local Awareness
>
> KNN+ computes the kth nearest neighbor distance between a test set embedding and the entire train set. This global approach doesn't account for the intrinsic cluster structure of the feature space. For example:
>
> Small k: Inter Cluster Density Variations. Consider a scenario with:
>
> •	Cluster A: Densely distributed
>
> •	Cluster B: Sparsely distributed
>
> Test observations belonging to cluster B will naturally be further from their corresponding training examples compared to those in A. Using a global threshold creates uniform-sized hyperspheres around test observations. This can lead to disproportionate OOD identification in sparse clusters like B compared to dense ones like A. We provide Toy Example 1 in the supplementary material (section 2) uploaded to our rebuttal revision response. We show two test set observations, one belonging to cluster A and one to B, their corresponding distances to the 5th nearest neighbor and the respective radii of the generated hyperspheres in 2D. The radius for cluster A is evidently smaller than that created for B. That means that given the level set estimation approach, more test observations from cluster B are likely to be closer to the 95% mark and identified as OOD.
>
> Large k: Intra Cluster Distance Variations. Consider a scenario with three clusters.
>
> •	Cluster C and D which are close to each other in the feature space
>
> •	Cluster E which is far from both C and D
>
> When k is large, distances from test observations might extend into neighboring cluster boundaries. This penalizes clusters that are further away and therefore well-separated from other clusters in the feature space. Paradoxically, this means better cluster separation could harm OOD detection performance. Please refer to Toy Example 2, which shows three test set observations, one belonging to cluster C, one to D and one to E, their corresponding distances to the 17th nearest neighbor and the radii of the generated hyperspheres in 2D.
>
> HAC's Advantage
>
> Our method addresses these limitations through distributional awareness. One could envision a KNN+ approach that incorporates class awareness, computing distances between test set and train set observations belonging to the same cluster and potentially normalized by class statistics. This approach would share some conceptual similarities with HAC.
>
> [1] Yiyou Sun, Yifei Ming, Xiaojin Zhu, and Yixuan Li. Out-of-distribution detection with deep nearest neighbors, 2022
>
> In response to your question about an ablation study examining the independent effect of radial and angular boundaries, we have produced two plots showing their interactions and effect on performance for CIFAR100 trained on ResNet34 (please refer to section 3 of the supplementary material). It's important to note that while angular and radial boundaries are parameters of each hypercone, they are not direct parameters of HAC. HAC uses the nearest neighbors (NNs) to select the angular boundary and a statistical measure to select the radial boundary. Therefore, our ablation study independently explores these two parameters.
> In the Figures, we see a general trend of better performance as NNs increases, stabilizing around 40 NNS and then slightly increasing. The dynamics of the increase vary across different preliminary boundary options. The graphs show how changes in NNs would affect FPR and AUROC performance for different preliminary boundaries. We conclude that while using the mean of ID lengths as the preliminary boundary of hypercones shows promising performance at small NNs, it deteriorates as NNs increases.
>
> Additionally, distribution aware boundaries (those which consider both the mean and standard deviation of ID lengths), tend to perform best and be more stable across varying NNs relative to percentile-based boundaries which may introduce sensitivity to outliers.
>
> 5. We will make these changes. Thank you!

---

> > ### Comment · Reviewer_KcXV · 2024-12-02
> >
> > Thank you for the response. The authors have addressed most of my concerns, and I appreciate the detailed explanations provided to clarify why the proposed method may outperform KNN. After considering other reviews, I have decided to maintain my original score.

---

### Official Review · Reviewer_CVgV · 2024-11-04

**Soundness:** 3
**Presentation:** 3
**Contribution:** 3
**Rating:** 5
**Confidence:** 3

**Summary:**

HACk-OOD is a post-training OOD detection method using hypercone projections to construct class-specific contours in embedding space. The approach achieves SOTA performance on CIFAR-100 and improves with larger networks. Including Imagenet experiments could further demonstrate scalability on large datasets.

**Strengths:**

HACk-OOD introduces a unique method using hypercone projections to delineate class contours, avoiding traditional Gaussian distribution assumptions and offering greater flexibility in complex feature spaces. The method achieves competitive, often superior, results on challenging datasets like CIFAR-100, demonstrating strong performance in both near and far OOD detection scenarios.

**Weaknesses:**

1. Experiments are limited to CIFAR-based datasets, testing on a large-scale dataset like Imagenet would better validate the method’s scalability. Also evaluating HACk-OOD on the OpenOOD benchmark would provide a clearer comparison to recent methods. I would consider rating this paper higher if Imagenet results were provided.

2. Missing comparisons with some of the latest post-hoc OOD methods, such as ASH and SCALE. Including these would offer a more comprehensive assessment of its relative performance.

Djurisic, Andrija, et al. "Extremely simple activation shaping for out-of-distribution detection." ICLR 2022
Xu, Kai, et al. "Scaling for training time and post-hoc out-of-distribution detection enhancement." ICLR 2023

**Questions:**

See Weakness.

---

> ### Author Response · Authors · 2024-11-25
>
> Thank you for all the insightful questions and for introducing us to two new OOD detection benchmarks. Please find our answers below:
>
> 1. Please refer to the rebuttal revision response on the Imagenet question. Regarding your question about the use of OpenOOD, we believe we have addressed the benchmark as best we could. With the exception of ImageNet, we have included training with in-distribution (ID) datasets such as CIFAR-10 and CIFAR-100, and tested against a broad range of out-of-distribution (OOD) datasets, including SVHN, Textures, Places365, as well as additional datasets such as iSUN and LSUN.
>
> 2. We have run ASH and SCALE and added the results to tables 1 and 2 of the manuscript. HAC still keeps its competitive advantage. The updated version of the paper, has been uploaded to OpenReview.

---

### Author Response · Authors · 2024-11-25

We would like to thank the reviewers for taking the time to read our manuscript and for their insightful questions. We responded to each individual question to the best of our ability. We hope that our responses are comprehensive and encourage the reviewers to ask follow-up questions if anything seems unclear.

Below we would like to tackle some of the repeating themes in questions raised by reviewers.

ImageNet Results:
We considered using ImageNet as an in-distribution (ID) dataset. However, due to legal considerations related to the indemnity provisions outlined in its terms of access, we are unable to do so. The terms state:
"Researcher accepts full responsibility for his or her use of the Database and shall defend and indemnify the ImageNet team, Princeton University, and Stanford University, including their employees, Trustees, officers, and agents, against any and all claims arising from Researcher's use of the Database, including but not limited to Researcher's use of any copies of copyrighted images that he or she may create from the Database."
Specifically, the lack of copyright information for images in the ImageNet dataset makes it incompatible with the data use policies to which we adhere, preventing us from accepting the terms of service. For this reason, we also might need to remove Imagenet-F from our Near-OOD experiments as well. We are still in discussions regarding our use of Imagenet-F and will provide an update as soon as we have one.
While we acknowledge that the absence of ImageNet results may be seen as a limitation, we believe the novelty and robustness demonstrated by our approach in the, harder, CIFAR-100 benchmark shows its potential effectively. We hope that our contributions be evaluated on the strength of our methodology and the results obtained using alternative datasets.

---

### Meta-Review · Area_Chair_crn5 · 2024-12-16

**Metareview:**

This paper introduces a novel post-hoc OOD detection approach that leverages hypercones to define class-specific boundaries in the embedding space. The method demonstrates a good performance on CIFAR-100 in both near-OOD and far-OOD scenarios. While the use of hypercones offers a unique perspective for OOD metric design and visualizations are provided in the supplementary material, there is no clear explanation or theoretical justification provided as to why the hypercones are superior for OOD detection. Additionally, the notation could be clearer. Key limitations include the lack of scalability testing on larger datasets like ImageNet, the absence of evaluations using modern architectures such as ViTs, and insufficient ablation studies to isolate the contributions of individual hypercone components as mentioned by multiple reviewers. Although the authors addressed many reviewer concerns with additional experiments and clarifications, the limited evaluation scope and unresolved questions resulted in a mixed reception, leaving the paper marginally below the threshold for acceptance.

**Additional Comments On Reviewer Discussion:**

- Multiple reviewers (CVgV, Fqvd) questioned the lack of experiments on larger and more realistic datasets such as ImageNet-1k and with ViT architectures (TiVv). The authors clarified that ImageNet results were unavailable due to copyright concerns and highlighted resource constraints. They extended evaluations with additional analyses and comparisons, but reviewers remained unconvinced.
- Some reviewers requested further clarity and theoretical analysis on why hypercones are advantageous for OOD detection compared to alternative methods and noted the similarity to Gaussian kernel-based methods. The authors provided detailed responses on the differences, emphasizing HACk-OOD’s focus on geometric boundaries rather than probabilistic modeling. While this was well-received, concerns about incremental novelty remained.
- Reviewers (KcXV, TiVv) requested more ablations to isolate the contributions of HACk-OOD’s angular and radial boundaries and independent contributions of hypercones. The authors added analyses in the supplementary materials. While informative, the lack of comparisons to alternative boundary methods (e.g., Gaussian mixtures) was seen as a gap.

The authors made efforts to address concerns, but reviewers maintained mixed ratings due to limited experimental validation and concerns about novelty and impact.

---

### Decision · Program_Chairs · 2025-01-22

Reject